# Sparse Modal Regression with Mode-Invariant Skew Noise

**Kazuki Koyama**  *kkoyama@ism.ac.jp*
*The Graduate University for Advanced Studies (SOKENDAI)*

**Takayuki Kawashima**  *kawashima@c.titech.ac.jp*
*Tokyo Institute of Technology*

**Hironori Fujisawa**  *fujisawa@ism.ac.jp*
*The Institute of Statistical Mathematics*
*The Graduate University for Advanced Studies (SOKENDAI)*
*RIKEN Center for Advanced Intelligence Project*

**Reviewed on OpenReview:** *https://openreview.net/forum?id=63r6M1JkXm*

## Abstract

Sparse regression methods have been widely used in many fields for their statistical effectiveness and high interpretability. However, there are few sparse regression methods with skew noise, although statistical modeling using skewness is becoming more important, e.g., in the medical field. The Azzalini's skew-normal distribution and its extensions are well-used for skew noise. Such skew regression methods have a severe problem with statistical interpretability because they model neither mean, median, nor mode. To overcome this problem, we propose a novel sparse regression method based on mode-invariant skew-normal noise. The regression model is easy to interpret in the proposed method because it always models a mode regardless of skewness. The proposed method is simple to implement and optimize, suggesting it is highly scalable to other machine-learning methods. We also provide theoretical guarantees of the proposed method for the average excess risk and the estimation error. Numerical experiments on artificial and real-world data demonstrate that the proposed method performs significantly better and is more stable than other existing methods for various skew-noise data.

## 1 Introduction

Starting with Lasso (Tibshirani, 1996), an $\ell_1$ regularization regression method is becoming popular for predictive modeling with relevant features. It has gained significant attention in many fields because of its ability to handle high-dimensional data efficiently and its exhaustive studies of methodological extensions and theoretical properties. (For details, e.g., see Bühlmann & Van De Geer (2011); Hastie et al. (2015).) As we have more data available and need more reasonable models, an $\ell_1$ regularization regression method will likely continue to be more essential for high-dimensional data analysis.

The Lasso is based on the squared error; in other words, it focuses on a symmetric noise. In real-world data analysis, however, we need to treat skewness. For example, statistical modeling using skewness is indispensable in the medical field. Hossain & Beyene (2015) explored an application of the skew-normal distribution in the analysis of microRNA (miRNA) data, which often does not follow a normal distribution. Their findings, derived from both simulations and real miRNA dataset analyses, demonstrated that the skew-normal distribution could enhance the detection of differentially expressed miRNAs. This enhancement is particularly noticeable when the data is significantly skewed. Shafiei et al. (2020) introduced an automated stain normalization framework for histopathology images that uses a mixture of multivariate skew-normal distributions to capture both symmetric and non-symmetric observations. The method can model multimodal and multiple correlated distributions, regardless of their non-symmetry, and has shown consistent and

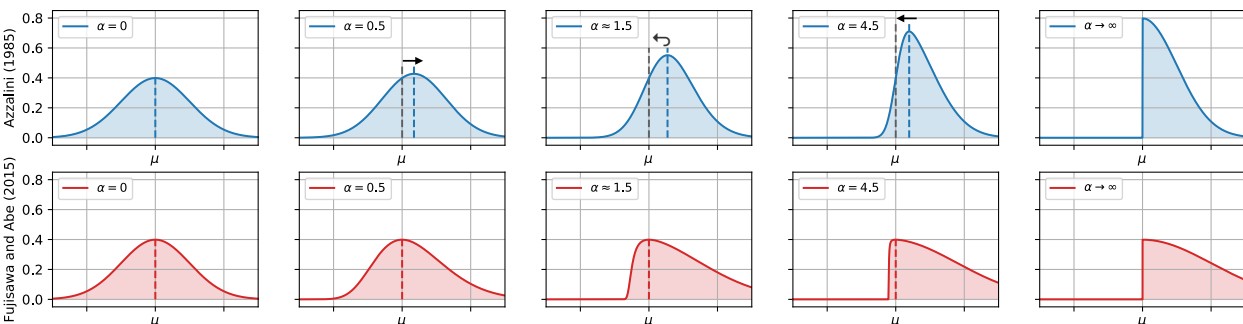

Figure 1: The difference between the Azzalini's skew-normal distribution (upper row) and the mode-invariant skew-normal distribution (lower row) with the location parameter $\mu$ and the different skewness parameter $\alpha$. The scale parameter is set to $\sigma = 1$. The blue and black dashed lines in the upper row represent the positions of the mode and $\mu$ in the former distribution, respectively. The mode of the former distribution depends complicatedly on the skewness parameter as well as the location and scale parameters. As $\alpha$ increases, the mode goes first to the right and then back to the left (to the original location). This behavior makes it difficult to interpret the former distribution. The red dashed line in the lower row represents the positions of both the mode and $\mu$ in the latter distribution. The mode of the latter distribution is invariant for any $\alpha$. As $\alpha \to \infty$, both asymptotically approach half-normal distributions. The former is stretched vertically, but the latter is stretched horizontally.

superior performance compared to existing methods. How to treat skewness is also a topic of interest in other fields, including financial, meteorological, and industrial data (Adcock et al., 2015; Hao et al., 2019; Simola et al., 2019).

Skew distributions have been studied extensively (Azzalini & Capitanio, 1999; Branco & Dey, 2001; Sahu et al., 2003; Liseo & Loperfido, 2003; González-Farías et al., 2004; Arellano-Valle & Genton, 2005; Arellano-Valle & Azzalini, 2006). However, despite the intense need for skewness in real-world data analysis, there currently exist few relevant $\ell_1$ regularization regression methods. The most significant study is Chen et al. (2014), who assumed a regression model for the location parameter of the well-used Azzalini's skew-normal distribution (Azzalini, 1985) and then proposed an $\ell_1$ regularization regression method with an efficient parameter estimation algorithm. Even if we move away from the $\ell_1$ regularization, there exist few regularization methods; e.g., Cao et al. (2023) recently proposed the ridge regularization regression method for the mode of the Azzalini's skew-normal distribution. This lack of attention may be because we would generally deal with skewed data using non-linear pre-processing with a power transform such as the Box-Cox transformation (Box & Cox, 1964) and Yeo-Johnson transformation (Yeo & Johnson, 2000). However, these non-linear transformations may not be suitable for a regression model with skewed noise because such transformations can provide symmetric distributions of outcomes but not symmetric noises due to asymmetric feature distributions. Furthermore, non-linear transformations can make it difficult to interpret a regression model since the non-linearity changes the meaning of an observation unit.

In this paper, we focus on a regression model whose noise is assumed to follow the mode-invariant skew-normal distribution introduced by Fujisawa & Abe (2015), and then we propose the regression estimator of the mode (location parameter) that is defined as the minimizer of the negative log-likelihood plus the $\ell_1$ regularization, aiming to achieve both skewness modeling and statistical interpretability. The mode-invariant skew-normal distribution is a subclass of the Transformation-of-Scale (ToS) distribution (Jones, 2014; 2016) and is consistent with a normal distribution when not skewed. As the name implies, its mode is invariant regardless of skewness, where its location parameter coincides with its mode. Therefore, the proposed method always models a mode regardless of skewness, making it easy to interpret the statistical role of sparse features. This is equivalent to the reasonable assumption that a noiseless situation has the highest probability. Despite its helpful properties, there are few studies with the mode-invariant skew-normal distribution besides the basic maximum likelihood estimation (Fujisawa & Abe, 2015).

The proposed method differs significantly from Chen et al. (2014)'s regression model for the location parameter of the Azzalini's skew-normal distribution. Figure 1 illustrates different shapes between the Azzalini's skew-normal distribution and the mode-invariant skew-normal distribution. Note that the location parameter of the Azzalini's skew-normal distribution is neither mean, median, nor mode, which means that the role of Chen et al. (2014)'s regression model is uncertain and difficult to interpret.

Our main contributions are as follows:

- Highlighting the attractiveness of the mode-invariant skew-normal distribution and assuming noise to follow it, we propose a novel $\ell_1$ regularization regression method having both skewness modeling and statistical interpretability.

- Revealing partial convexity, we indicate that its optimization and implementation are relatively straightforward, although the proposed method involves complicated non-linear transformations.

- The proposed method is theoretically guaranteed in terms of its excess risk and estimation error by non-asymptotic analysis under mild assumptions usually adopted in the theoretical analysis of Lasso-type problems.

This paper is organized as follows. In Section 2, the proposed method is introduced. In Section 3, the theoretical guarantees of the proposed method are provided by non-asymptotic analysis in terms of the excess risk and the estimation error. Such theoretical studies for skew regression models have not been shown so far. In Section 4, numerical experiments on artificial and real-world data are demonstrated. The results obtained in that study indicate that the proposed method can serve as a viable alternative to Lasso. In Section 5, concluding remarks are given.

## 2 Proposed Method

### 2.1 Skew Distribution

We use the mode-invariant skew-normal distribution (Fujisawa & Abe, 2015) with the parameter vector $\boldsymbol{\psi} := [\mu, \sigma, \alpha]^\top \in \mathbb{R} \times \mathbb{R}_+ \times \mathbb{R}$:

$$f(y \mid \boldsymbol{\psi}) := \frac{1}{\sigma} \phi \left( r_\alpha \left( \frac{y - \mu}{\sigma} \right) \right), \tag{1}$$

where $\phi$ is the probability density function of the standard normal distribution, $r_\alpha$ is a special transformation function of scale, described later, and $\mu$, $\sigma$, and $\alpha$ are the parameters of location, scale, and skewness, respectively. The function $r_\alpha$ must satisfy

$$r'_\alpha(u) > 0, \tag{2}$$
$$q'_\alpha(v) + q'_\alpha(-v) = 2, \tag{3}$$

where $q_\alpha$ is the inverse function of $r_\alpha$. Note that the condition (2) ensures a monotone-increasing property of $r_\alpha(u)$. The model (1) is unimodal with the mode-invariant property $\mathrm{Mode}\left[f(y \mid \boldsymbol{\psi})\right] = \mu$ for any $\sigma$ and $\alpha$.

Let $H_\alpha(v) := q_\alpha(v) - v$. While $H_\alpha$ (or $r_\alpha$) can be freely designed if it satisfies the above conditions, this paper employs the following special function $H_\alpha$ because of its better analytical treatment:

$$H_\alpha(v) := \rho_\alpha \frac{\sqrt{1 + \alpha^2 v^2} - 1}{\alpha}, \tag{4}$$

where $\rho_\alpha := 1 - \frac{1}{2} \exp(-\alpha^2)$.

**Remark 1.** *The above function $\rho_\alpha$ slightly differs from that proposed in Fujisawa & Abe (2015) because the Fisher information matrix is non-singular at $\alpha = 0$ when we use the above function but singular when we use the function proposed in Fujisawa & Abe (2015). The non-singularity is usually necessary to obtain some theoretical properties. A related proposition appears in Proposition 2.*

## 2.2 Problem Description

We consider the linear regression model with the skew-unimodal noise, given by

$$y = \boldsymbol{X}^\top \boldsymbol{\beta} + e, \tag{5}$$

where $y \in \mathbb{R}$ is an output, $\boldsymbol{X} \in \mathbb{R}^P$ is a $P$-dimensional feature vector, $\boldsymbol{\beta} \in \mathbb{R}^P$ is a parameter vector, and $e$ follows a skew-unimodal distribution $f(e \mid 0, \sigma, \alpha)$. Let $\boldsymbol{\psi}(\boldsymbol{X}) := [\boldsymbol{X}^\top \boldsymbol{\beta}, \sigma, \alpha]^\top$. The distribution of $y$ can be expressed by $f(y \mid \boldsymbol{\psi}(\boldsymbol{X}))$, and the mode is $\boldsymbol{X}^\top \boldsymbol{\beta}$.

Let the log-likelihood function be denoted by $l(\boldsymbol{\theta} \mid \boldsymbol{X}, y) := \log f(y \mid \boldsymbol{\psi}(\boldsymbol{X}))$ with the parameter vector $\boldsymbol{\theta} := [\boldsymbol{\beta}^\top, \sigma, \alpha]^\top$. Let $\mathcal{D}_N := \{(y_n, \boldsymbol{X}_n)\}_{n=1}^N$ be the i.i.d. data. We define the loss function by

$$\ell(\boldsymbol{\theta} \mid \mathcal{D}_N) := -\frac{1}{N} \sum_{n=1}^N l(\boldsymbol{\theta} \mid \boldsymbol{X}_n, y_n)$$

$$= -\frac{1}{N} \sum_{n=1}^N \log f(y_n \mid \boldsymbol{\psi}(\boldsymbol{X}_n)). \tag{6}$$

Since we are interested in sparse regression, this paper focuses on the Lasso-type problem

$$\min_{\boldsymbol{\theta}} \ell(\boldsymbol{\theta} \mid \mathcal{D}_N) + \lambda \|\boldsymbol{\beta}\|_1, \tag{7}$$

where $\lambda \geq 0$ is the regularization parameter.

## 2.3 Optimization

The loss function (6) is non-convex for $\boldsymbol{\theta}$ but convex for $\boldsymbol{\beta}$ when $\sigma$ and $\alpha$ are fixed, as described in the following theorem.

**Theorem 1.** *The loss function $\ell(\boldsymbol{\theta} \mid \mathcal{D}_N)$ is convex with a Lipschitz continuous gradient for $\boldsymbol{\beta}$. In particular, if $S_N := \frac{1}{N} \sum_{n=1}^N \boldsymbol{X}_n \boldsymbol{X}_n^\top$ is positive definite, $\ell(\boldsymbol{\theta} \mid \mathcal{D}_N)$ is strongly convex for $\boldsymbol{\beta}$.*

The proof is given in Appendix B. Using this theorem, we develop a problem-specific optimization method based on the block coordinate descent method. This method allows us to separate the optimization problem into convex and non-convex and take advantage of the convenient property described in Theorem 1.

The update algorithm is summarized in Algorithm 1. We update $\boldsymbol{\beta}$, $\sigma$, and $\alpha$ alternately. First, we fix $(\sigma, \alpha)$ and optimize $\boldsymbol{\beta}$. We employ the Majorization-Minimization (MM) algorithm using the Taylor expansion with acceleration techniques (Jamshidian & Jennrich, 1997; Sun et al., 2016). The derivation of this MM algorithm is given in Appendix C. As shown in line 8 of Algorithm 1, we can rewrite the $\boldsymbol{\beta}$ update as another Lasso-type problem and then use well-known software, e.g., the *sklearn.linear_model* package of Python. Next, we fix $(\boldsymbol{\beta}, \alpha)$ and optimize $\sigma$, and finally fix $(\boldsymbol{\beta}, \sigma)$ and optimize $\alpha$. Although we need to consider the non-convexity for $\sigma$ and $\alpha$, their optimization problems are at most one variable. Hence, we can easily use trustable software, e.g. the *scipy.optimize* package of Python. In this paper, we employ the L-BFGS algorithm (Liu & Nocedal, 1989), which is an iterative method for solving non-linear optimization problems. In particular, for the inequality constraint $\sigma > 0$, we can utilize the L-BFGS-B algorithm (Byrd et al., 1995; Zhu et al., 1997), which extends the L-BFGS algorithm to handle bounded constraints.

Let $\tilde{\ell}(\boldsymbol{\theta}) := \ell(\boldsymbol{\theta} \mid \mathcal{D}_N) + \lambda \|\boldsymbol{\beta}\|_1$. Let dom $\tilde{\ell}$ be the effective domain of $\tilde{\ell}$. Following the definition of Tseng (2001), we say that $\boldsymbol{\theta} \in \text{dom } \tilde{\ell}$ is a stationary point of $\tilde{\ell}$ if for any $\boldsymbol{d} \in \mathbb{R}^{P+2}$,

$$\liminf_{a \downarrow 0} \frac{\tilde{\ell}(\boldsymbol{\theta} + a\boldsymbol{d}) - \tilde{\ell}(\boldsymbol{\theta})}{a} \geq 0. \tag{8}$$

Algorithm 1 guarantees convergence to a stationary point, as described in the following theorem.

**Theorem 2.** *Let $\{\boldsymbol{\theta}^{(t)}\}_{t=0,1,\ldots}$ be a sequence of $\boldsymbol{\theta}$ after updating $\sigma$ in Algorithm 1. Then, every cluster point of $\{\boldsymbol{\theta}^{(t)}\}_{t=0,1,\ldots}$ is a stationary point of $\tilde{\ell}$.*

---

**Algorithm 1** Optimization of the proposed method

---
**Require:** Hyper-parameter $\lambda$ and initialized $\boldsymbol{\beta}$, $\sigma$, $\alpha$

1: **while** until convergence **do**
2:    **while** until convergence (of $\boldsymbol{\beta}$) **do**
3:       **for** $n = 1, \ldots, N$ **do**
4:          $z_n \leftarrow \frac{\alpha}{\sigma} \left( y_n - \boldsymbol{X}_n^\top \boldsymbol{\beta} \right) + \rho_\alpha$
5:          $w_n \leftarrow \sqrt{z_n^2 + 1 - \rho_\alpha^2}$
6:          $\tilde{y}_n \leftarrow \boldsymbol{X}_n^\top \boldsymbol{\beta} + \frac{\sigma}{2\alpha} \left( z_n - \frac{\rho_\alpha}{1+\rho_\alpha^2} \left( w_n + \frac{z_n^2}{w_n} \right) \right)$
7:       **end for**
8:       $\boldsymbol{\beta} \leftarrow \underset{\boldsymbol{\beta}}{\operatorname{argmin}} \frac{1+\rho_\alpha^2}{N\sigma^2(1-\rho_\alpha^2)^2} \sum_{n=1}^N \left( \tilde{y}_n - \boldsymbol{X}_n^\top \boldsymbol{\beta} \right)^2 + \lambda \|\boldsymbol{\beta}\|_1$
9:    **end while**
10:   $\sigma \leftarrow \underset{\sigma:\, \sigma > 0}{\operatorname{argmin}} \ell(\boldsymbol{\theta} \mid \mathcal{D}_N)$ with L-BFGS-B (or other valid algorithm)
11:   $\alpha \leftarrow \underset{\alpha}{\operatorname{argmin}} \ell(\boldsymbol{\theta} \mid \mathcal{D}_N)$ with L-BFGS (or other valid algorithm)
12: **end while**

---

The proof is given in Appendix D. Here, we describe how to set initial values in this paper. The regression parameter $\boldsymbol{\beta}$ started from the estimated coefficients of the Lasso model for the same data. Using the residuals of its model, we initialized $\sigma$ with their sample standard deviation and $\alpha$ with $-1$ or $1$ corresponding to the sign of their sample skewness.

### 2.4 Related Work

The most famous skew distribution is the Azzalini's skew-normal distribution (Azzalini, 1985), implemented in current standard scientific computing software. Many skew distributions were developed using the Azzalini's basic idea (Azzalini & Capitanio, 1999; Branco & Dey, 2001; Sahu et al., 2003; Liseo & Loperfido, 2003; González-Farías et al., 2004; Arellano-Valle & Genton, 2005; Arellano-Valle & Azzalini, 2006). The closest method to the proposed method is Chen et al. (2014), which assumes a regression model for the location parameter of the Azzalini's skew-normal distribution. However, the location parameter in these skew distributions is neither mean, median, nor mode; the mean, median, and mode have complicated combinations of all the location, scale, and skewness parameters. Even if the location is modeled by $\boldsymbol{X}^\top \boldsymbol{\beta}$ for the Azzalini's skew-normal distribution, it is difficult to understand the role of features. The proposed modeling makes it easy to understand the role of features via the modeling of mode. The proposed modeling is suitable for an interpretable one in a high-dimensional setting, but the modelings using the Azzalini's skew-normal distribution are not.

## 3 Theoretical Results

This section provides theoretical guarantees for the proposed method by non-asymptotic analysis. First, we present some basic properties of the first, second, and third derivatives of $\ell(\boldsymbol{\psi} \mid y) = \log f(y \mid \boldsymbol{\psi})$. Next, using them, we show that the excess risk has a quadratic margin. Finally, we show the upper bounds of the average excess risk and the estimation error, and then we verify the favorable properties of the proposed method. All proofs of propositions and theorems in this section are given in Appendices E, F, G, and H.

### 3.1 Basic Properties of Likelihood

We present some basic properties of the first, second, and third derivatives of $\ell(\boldsymbol{\psi} \mid y)$. These properties are the basis of non-asymptotic analysis.

Suppose that $\mathcal{X} = \{\boldsymbol{X}\}$ is included in a bounded space of $\mathbb{R}^P$. We assume that the parameter spaces $\Psi = \{\boldsymbol{\psi}\}$ and $\Theta = \{\boldsymbol{\theta}\}$ are bounded in the following sense. Hereafter, we suppose that the following assumption is satisfied.

**Assumption 1.** *For some positive constant $K$, we suppose*

$$\Psi \subset \tilde{\Psi} = \{\boldsymbol{\psi} : |\mu| \leq K, \ 1/K \leq |\sigma| \leq K, \ |\alpha| \leq K\}, \tag{9}$$

$$\Theta \subset \tilde{\Theta} = \{\boldsymbol{\theta} : \boldsymbol{\psi}(\boldsymbol{X}) \in \tilde{\Psi} \text{ for any } \boldsymbol{X} \in \mathcal{X}\}. \tag{10}$$

*The true parameters $\boldsymbol{\psi}^0$ and $\boldsymbol{\theta}^0$ are included in $\Psi$ and $\Theta$, respectively.*

Let the score function (first derivative) of $l(\boldsymbol{\psi} \mid y) := \log f(y \mid \boldsymbol{\psi})$ be denoted by

$$s_{\boldsymbol{\psi}}(y) := \frac{\partial}{\partial \boldsymbol{\psi}} l(\boldsymbol{\psi} \mid y). \tag{11}$$

**Proposition 1.** *We have*

$$\|s_{\boldsymbol{\psi}}(y)\|_\infty \leq C_{1,2}|y|^2 + C_{1,1}|y| + C_{1,0}, \tag{12}$$

*where $C_{1,2}, C_{1,1}, C_{1,0}$ are some positive constants.*

The score functions of regression parameters $\mu$ and $\sigma$ are bounded by the first and second-order polynomials of $|y|$. This is the same property as the normal linear regression model. We may expect that the score function of skew parameter $\alpha$ is bounded by a third-order polynomial of $|y|$ because the skewness usually depends on the third moment. However, it is bounded by a second-order polynomial of $|y|$. Proposition 1 implies that the proposed model is easy to treat $\alpha$ as well as $\sigma$.

Let the Fisher information matrix of $f_{\boldsymbol{\psi}}(y) := f(y \mid \boldsymbol{\psi})$ be denoted by

$$\mathcal{I}(\boldsymbol{\psi}) := \mathbb{E}_{f_{\boldsymbol{\psi}}} \left[ -\frac{\partial^2}{\partial \boldsymbol{\psi} \partial \boldsymbol{\psi}^\top} l(\boldsymbol{\psi} \mid y) \right]. \tag{13}$$

**Proposition 2.** *Let $\boldsymbol{\psi}^0(\boldsymbol{X}) := [\boldsymbol{X}^\top \boldsymbol{\beta}^0, \sigma^0, \alpha^0]^\top$. The Fisher information matrix $\mathcal{I}(\boldsymbol{\psi}^0(\boldsymbol{X}))$ is uniformly positive definite for any $\boldsymbol{X}$, in other words, $\inf_{\boldsymbol{X}} \Lambda_{\min}(\mathcal{I}(\boldsymbol{\psi}^0(\boldsymbol{X}))) > 0$, where $\Lambda_{\min}(A)$ is the smallest eigenvalue of $A$.*

If the model is simple, we can easily prove a similar proposition to Proposition 2 because the Fisher information matrix is easily calculated. However, the model (1) is not simple, and it is not easy to verify Proposition 2, as seen in a devised proof of Appendix E.3. We found that the constant factor $\rho_\alpha$ proposed by Fujisawa & Abe (2015) does not satisfy Proposition 2 at $\alpha = 0$. This is why a constant factor $\rho_\alpha$ is slightly modified, as described in Remark 1, and then we can prove Proposition 2.

**Proposition 3.** *There exists some function $G_3(y)$ such that*

$$\sup_{(i_1,i_2,i_3) \in \{1,2,3\}^3} \left| \frac{\partial^3}{\partial \psi_{i_1} \partial \psi_{i_2} \partial \psi_{i_3}} l(\boldsymbol{\psi} \mid y) \right| \leq G_3(y),$$

$$C_3 := \sup_{\boldsymbol{X}} \int G_3(y) f(y \mid \boldsymbol{\psi}^0(\boldsymbol{X})) dy < \infty. \tag{14}$$

## 3.2 Excess Risk and Margin

Let the Kullback-Leibler divergence between $f_{\boldsymbol{\psi}^0}$ and $f_{\boldsymbol{\psi}}$, which is called the *excess risk*, be denoted by

$$\mathcal{E}(\boldsymbol{\psi} \mid \boldsymbol{\psi}^0) := \mathbb{E}_{f_{\boldsymbol{\psi}^0}} \left[ \log \frac{f_{\boldsymbol{\psi}^0}(y)}{f_{\boldsymbol{\psi}}(y)} \right]. \tag{15}$$

**Proposition 4.** *For any $\epsilon > 0$, there exists some constant $\delta_\epsilon > 0$ such that*

$$\inf_{\boldsymbol{X}} \inf_{\|\boldsymbol{\psi} - \boldsymbol{\psi}^0(\boldsymbol{X})\|_2 > \epsilon} \mathcal{E}(\boldsymbol{\psi} \mid \boldsymbol{\psi}^0(\boldsymbol{X})) \geq \delta_\epsilon. \tag{16}$$

Proposition 4 shows the *identifiability condition*, which is usually necessary to estimate the true parameter by a statistical method correctly. The identifiability condition is often assumed implicitly, but it is precisely verified here because the model (1) is complicated due to skew noise.

**Theorem 3.** *There exists some constant $C > 0$ such that*

$$\inf_{\boldsymbol{X}} \frac{\mathcal{E}\left(\boldsymbol{\psi} \mid \boldsymbol{\psi}^0(\boldsymbol{X})\right)}{\|\boldsymbol{\psi} - \boldsymbol{\psi}^0(\boldsymbol{X})\|_2^2} \geq \frac{1}{C^2}. \tag{17}$$

Theorem 3 says that the proposed method has a *quadratic margin*, implying the error $\mathcal{E}\left(\boldsymbol{\psi} \mid \boldsymbol{\psi}^0(\boldsymbol{X})\right)$ presents a sharper behavior around $\boldsymbol{\psi}^0(\boldsymbol{X})$ than the well-used squared error $\|\boldsymbol{\psi} - \boldsymbol{\psi}^0(\boldsymbol{X})\|_2^2$ up to a constant factor.

### 3.3 Convergence Rate

The following is a well-used condition in non-asymptotic analysis for the $\ell_1$ penalized method, which is called the *restricted eigenvalue condition*.

**Assumption 2.** *Let $\mathcal{S}_0 := \{p : \beta_p^0 \neq 0\}$ and $\mathcal{S}_0^C := \{1, \ldots, P\} \backslash \mathcal{S}_0$. With $L > 1$, define the $(L, \mathcal{S}_0)$-restricted eigenvalue constant by*

$$\tilde{\kappa}^2(L, \mathcal{S}_0) := \inf_{\|\boldsymbol{\beta}_{\mathcal{S}_0^C}\|_1 \leq L \|\boldsymbol{\beta}_{\mathcal{S}_0}\|_1} \frac{\boldsymbol{\beta}^\top S_N \boldsymbol{\beta}}{\|\boldsymbol{\beta}\|_2^2}. \tag{18}$$

*We set $L = 6$ and assume $0 < \tilde{\kappa}(L, \mathcal{S}_0) \leq 1$.*

The squared loss of the simple linear regression model for a low-dimensional case is strongly convex because the Hessian matrix of the squared loss concerning the regression parameter vector $\boldsymbol{\beta}$ is $S_N$ and it is positive definite. This property provides the strength of the least squared estimator. However, for the high-dimensional case $N < P$, $S_N$ may not be of full rank; in other words, its smallest eigenvalue may be zero, and then the squared loss is not strongly convex. The non-asymptotic analysis reveals that if the smallest eigenvalue of $S_N$ under the *restricted condition* $\|\boldsymbol{\beta}_{\mathcal{S}_0^C}\|_1 \leq L \|\boldsymbol{\beta}_{\mathcal{S}_0}\|_1$ is positive, such as Assumption 2, we can show theoretical guarantees of the $\ell_1$ regularized estimator (Bühlmann & Van De Geer, 2011; Hastie et al., 2015).

In Assumption 2, we set $L = 6$ to simplify later notations. We can replace it with any constant greater than one by adjusting how to choose the regularization parameter. Based on this assumption, we have the following important theorem.

**Theorem 4.** *Suppose that Assumption 2 is satisfied. Fix $\boldsymbol{X}_1, \ldots, \boldsymbol{X}_N$. Let the average excess risk and the empirical process of the loss function* (6) *be denoted by*

$$\bar{\mathcal{E}}(\boldsymbol{\theta} \mid \boldsymbol{\theta}^0) := \frac{1}{N} \sum_{n=1}^{N} \mathcal{E}\left(\boldsymbol{\psi}(\boldsymbol{X}_n) \mid \boldsymbol{\psi}^0(\boldsymbol{X}_n)\right), \tag{19}$$

$$\mathcal{V}_N(\boldsymbol{\theta}) := \frac{1}{N} \sum_{n=1}^{N} \left(l\left(\boldsymbol{\psi}(\boldsymbol{X}_n) \mid y_n\right) - \mathbb{E}\left[l\left(\boldsymbol{\psi}(\boldsymbol{X}_n) \mid y\right)\right]\right). \tag{20}$$

*Let $\|\boldsymbol{\theta}\|_* := \|\boldsymbol{\beta}\|_1 + \|[\sigma, \alpha]^\top\|_2$. Fix $\gamma \geq 1$ and $\lambda_0 > 0$. Let*

$$\mathcal{J} := \left\{ \frac{|\mathcal{V}_N(\boldsymbol{\theta}) - \mathcal{V}_N(\boldsymbol{\theta}^0)|}{\|\hat{\boldsymbol{\theta}} - \boldsymbol{\theta}^0\|_* \vee \lambda_0} \leq \gamma \lambda_0 \right\}. \tag{21}$$

*If the regularization parameter $\lambda$ is chosen to satisfy $\lambda \geq 2\gamma\lambda_0$, then we have on $\mathcal{J}$*

$$\bar{\mathcal{E}}(\hat{\boldsymbol{\theta}} \mid \boldsymbol{\theta}^0) + 2(\lambda - \gamma\lambda_0)\|\hat{\boldsymbol{\beta}} - \boldsymbol{\beta}^0\|_1 \leq 9(\lambda + \gamma\lambda_0)^2 C^2 \kappa^2 s_0, \tag{22}$$

*where $C$ is the constant defined in Theorem 3, $\kappa^{-1} := \tilde{\kappa}(L, \mathcal{S}_0)$, and $s_0 := \#\mathcal{S}_0$.*

Suppose

$$\lambda_0 = O\left((\log N)^2 \sqrt{\frac{\log(N \vee P)}{N}}\right). \tag{23}$$

This order is necessary to prove Theorem 5 and is relevant to the following two corollaries. From Theorem 4, we can obtain convergence rates regarding the average excess risk $\bar{\mathcal{E}}(\hat{\boldsymbol{\theta}} \mid \boldsymbol{\theta}^0)$ and the estimation error of $\hat{\boldsymbol{\beta}}$.

**Corollary 1.** *Suppose $\lambda = 2\gamma\lambda_0$ with* (23) *and the same assumptions as in Theorem 4. Then, we have*

$$\bar{\mathcal{E}}(\hat{\boldsymbol{\theta}} \mid \boldsymbol{\theta}^0) = O_{\mathbb{P}}\left(\frac{s_0(\log N)^4 \log(N \vee P)}{N}\right), \tag{24}$$

$$\|\hat{\boldsymbol{\beta}} - \boldsymbol{\beta}^0\|_1 = O_{\mathbb{P}}\left(s_0(\log N)^2 \sqrt{\frac{\log(N \vee P))}{N}}\right). \tag{25}$$

The orders of the average excess risk (24) and the estimation error (25) are $(\log N)^4$ and $(\log N)^2$ larger than those of the ordinary Lasso, respectively. These are the same as in some well-known complicated models, such as the linear mixed effects models with $\ell_1$ regularization (Schelldorfer et al., 2011). A similar result is also seen in Corollary 2.

### 3.4 Feature Selection

When the non-zero coefficients are sufficiently large, which is the so-called beta-min condition (Bühlmann, 2012), we can show that the estimated set of non-zero coefficients $\hat{\mathcal{S}} := \{p : \hat{\beta}_p \neq 0\}$ includes the true set of non-zero coefficients $\mathcal{S}_0$ with high probability.

**Corollary 2.** *Suppose $\lambda = 2\gamma\lambda_0$ with* (23) *and the same assumptions as in Theorem 4. Suppose the non-zero coefficients of $\boldsymbol{\beta}_0$ are sufficiently large so that*

$$\min_{p \in \mathcal{S}_0} |\beta_p^0| \gg O\left(s_0(\log N)^2 \sqrt{\frac{\log(N \vee P))}{N}}\right). \tag{26}$$

*Then, with high probability, it holds $\hat{\mathcal{S}} \supseteq \mathcal{S}_0$.*

As in Corollary 1, the order in (26) is $(\log N)^2$ larger when compared to the ordinary Lasso.

### 3.5 Reasonable Success Probability

The statements after Section 3.3 hold when the event $\mathcal{J}$ in (21) occurs. Finally, we show that the event $\mathcal{J}$ occurs with high probability.

**Theorem 5.** *Suppose $\lambda = 2\gamma\lambda_0$ with* (23) *and the same assumptions as in Theorem 4. The event $\mathcal{J}$ occurs with probability at least $1 - \delta$, where with some positive constants $C_{4,1}, C_{4,2} > 0$,*

$$\delta := C_{4,1} \exp\left(-\frac{\gamma^2(\log N)^2 \log(N \vee P)}{C_{4,2}^2}\right) + \frac{1}{N}. \tag{27}$$

## 4 Experimental Results

This section demonstrates the performance of the proposed method with synthetic and real-world datasets.

### 4.1 Skew-Noises with Mode Zero

The simulation model is $y = \boldsymbol{X}^\top \boldsymbol{\beta}^0 + e$. The feature vector $\boldsymbol{X}_n \in \mathbb{R}^{P=100}$ was generated randomly with each element following the continuous uniform distribution on $[-1, 1] \in \mathbb{R}$. The true parameter was set

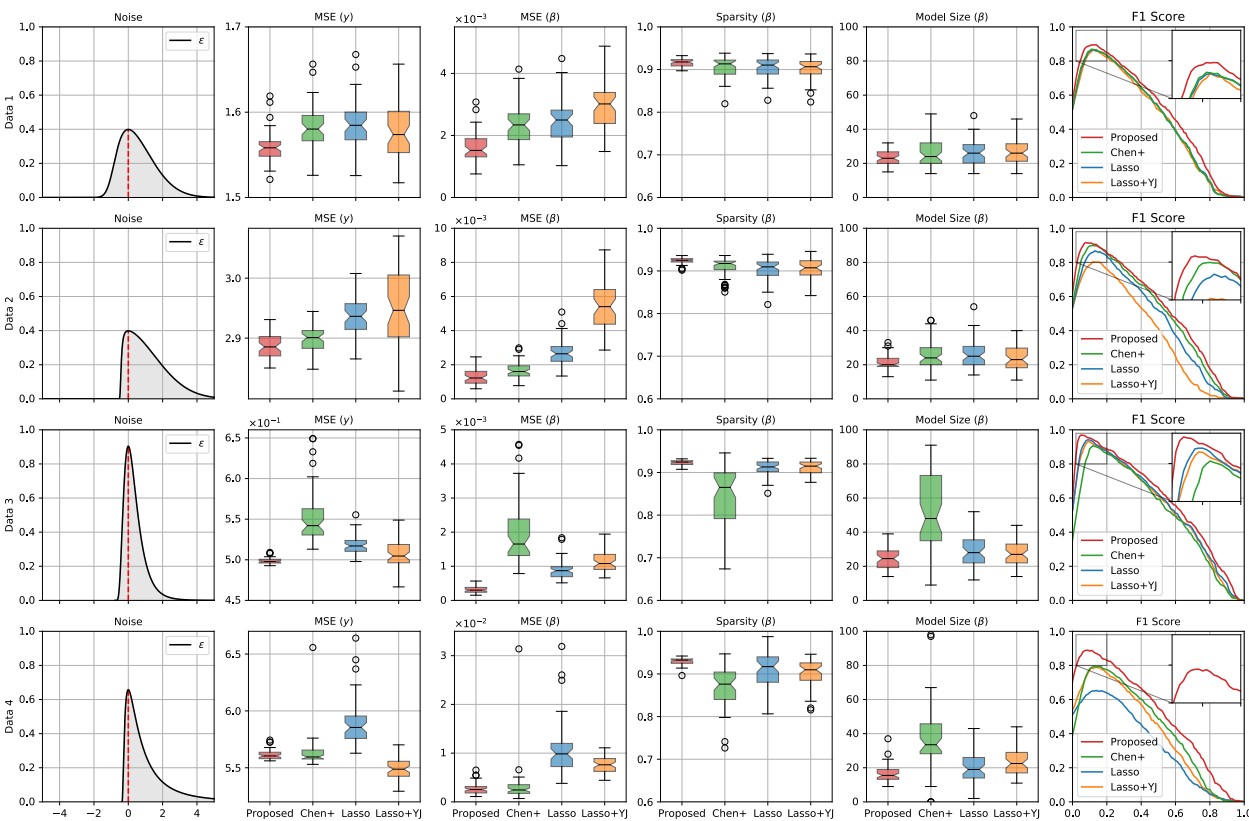

Figure 2: Comparison of four methods with four different types of simulation noises in terms of five evaluation measures. From top to bottom, each row results from Data 1, Data 2, Data 3, and Data 4, respectively. The first column plots the probability density function of the noise. The second through fifth columns are box plots of some scores. The last column is the mean F1 score for each threshold (horizontal axis). In all figures, the proposed method, Chen+ (Chen et al., 2014), Lasso, and Lasso with the Yeo-Johnson transformation are in red, green, blue, and yellow, respectively. All experiments were conducted for 50 runs with different random seeds.

to $\boldsymbol{\beta}^0 = [1.0, -0.9, 0.8, ..., -0.1, 0, ..., 0]^\top \in \mathbb{R}^{100}$ with 10% non-zero coefficients of varying magnitudes and signs. Let the random variable following (1) be denoted by $\mathcal{SN}(\mu, \sigma, \alpha)$. Let the log-normal variable be denoted by $\ln\mathcal{N}(\mu, \sigma)$. The noise $e$ was generated from four types of distribution: $\mathcal{SN}(0, 1, 1)$ (**Data 1**), $\mathcal{SN}(0, 1, 2)$ (**Data 2**), $\ln\mathcal{N}(0, 0.5) - \exp(-0.5^2)$ (**Data 3**), and $\ln\mathcal{N}(0, 1) - \exp(-1^2)$ (**Data 4**). Data 3 and Data 4 are the log-normal variable adjusted to mode zero. These are misspecified models. The noise distributions are depicted in the first column of Figure 2.

We compared the proposed method with the ordinary Lasso (Tibshirani, 1996) ("Lasso"), the ordinary Lasso applied to the Yeo-Johnson transformed output ("Lasso+YJ"), and another skew-noise Lasso (Chen et al., 2014) ("Chen+") that assumes a regression model for the location parameter of the Azzalini's skew-normal distribution. The tuning power parameter in the Yeo-Johnson transformation was determined by maximum likelihood estimation. The sample size was set to $N = 500$. We conducted 50 experiments with different random seeds. For each trial, the regularization coefficient $\lambda$ was adjusted by 5-fold cross-validation based on the log-likelihood loss, in which the numbers of training and validation data for each trial were set to 400 and 100, respectively.

Five evaluation measures are shown in the second through six columns of Figure 2. The second column is the Mean Squared Error (MSE) for the prediction: $\text{MSE}(\hat{y}) := \sum_{n:\text{test}} (\hat{y}_n - y_n^0)^2 / N$, where the MSE was

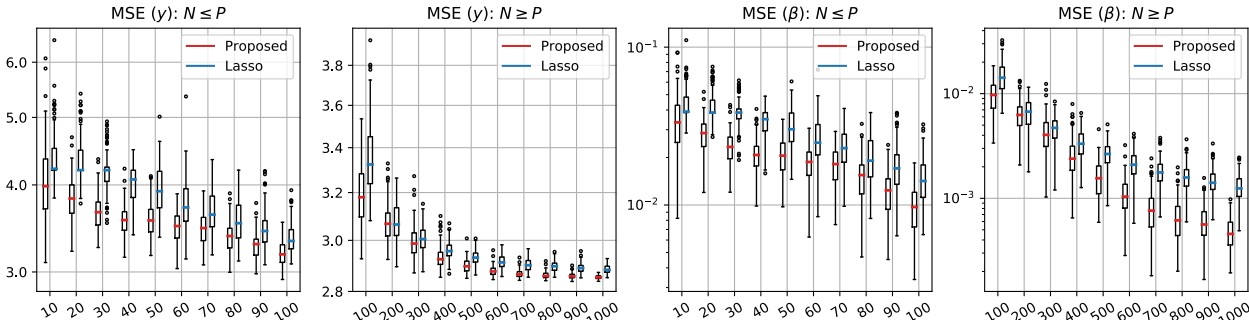

Figure 3: Box plots of MSEs for Data 2 with $P = 100$ fixed and various sample sizes $N$. The vertical axis is a logarithmic scale. The horizontal axis is the sample size $N$. All experiments were conducted for 100 runs with different random seeds.

estimated from 2,000 test data generated under the same conditions. The third column is the MSE of the estimated coefficients: $\mathrm{MSE}(\hat{\boldsymbol{\beta}}) := \|\hat{\boldsymbol{\beta}} - \boldsymbol{\beta}^0\|_2^2 / P$. Since "Lasso+YJ" is based on the Yeo-Johnson transformed output, the resulting estimator $\hat{\boldsymbol{\beta}}$ does not generally converge to $\boldsymbol{\beta}^0$. Therefore, $\mathrm{MSE}(\hat{\boldsymbol{\beta}})$ is expected to be inferior to other methods. The fourth column is a measure for sparsity by Hurley & Rickard (2009) as the most robust measure founded on a weighted sum of estimated all coefficients for evaluating sparsity. The measure range is $[0, 1] \in \mathbb{R}$; as a coefficient vector is sparser, its value is higher (see Appendix I.1 for details). The fifth column is the size of the estimated model, which is defined as the number of non-zero coefficients: $\mathrm{Model\,Size}(\hat{\boldsymbol{\beta}}) := \#\{\hat{\beta}_p \neq 0 : 1 \leq p \leq P\}$. The last column is the F1 score (Van Rijsbergen, 2004) defined by

$$\mathrm{F1\ Score} := 2 \times \frac{\mathrm{Precision} \times \mathrm{Recall}}{\mathrm{Precision} + \mathrm{Recall}}, \tag{28}$$

where Precision and Recall are defined by $\mathrm{TP}/(\mathrm{TP} + \mathrm{FP})$ and $\mathrm{TP}/(\mathrm{TP} + \mathrm{FN})$, respectively, and TP, FP, and FN mean the true positive, false positive, and false negative, respectively. The positivity (negativity) of the estimated coefficient is defined by $|\hat{\beta}_p|/\|\hat{\boldsymbol{\beta}}\|_\infty > \delta_T$ for a threshold $\delta_T \in [0, 1]$ (horizontal axis). As the feature selection is more successful at each threshold, the F1 score is larger. This means that a better method presents an upper curve.

As a result of Figure 2, the proposed method outperformed the comparative methods in all the cases, indicating better prediction and feature selection with a smaller number of features. We see that Chen+ is inferior to Lasso in Data 3. This might be because the Azzalini's skew-normal distribution could not adequately model such skewed noise.

Figure 3 shows box plots of MSEs for 100 trials regarding Data 2 with $P = 100$ fixed and various sample sizes $N$. The regularization coefficient $\lambda$ was tuned by 5-fold cross-validation with 80% training and 20% validation data. The Lasso is a benchmark against the proposed method. The proposed method presented stable behaviors for $N < P$ as well as $N > P$.

In Appendix I.2, we also show the results with a linear regression model without regularization, assuming the Azzalini's skew-normal distribution, skew-$t$ distribution, and skew-Cauchy distribution for noise (Arellano-Valle & Azzalini, 2013; Azzalini & Arellano-Valle, 2013; Azzalini & Salehi, 2020), on Data 1 through Data 4.

## 4.2   Normal and Azzalini's Skew-Noises

We explore both normal and Azzalini's skew-normal noises under the same conditions as in Section 4.1. Let the random variable following the Azzalini's skew-normal be denoted by $\mathcal{SN}_{\mathrm{A}}(\mu, \sigma, \alpha)$. The noise $e$ was generated from additional four types of distribution: $\mathcal{N}(0, 1)$ (**Data 5**), $\mathcal{SN}_{\mathrm{A}}(0, 1, 2)$ (**Data 6**), $\mathcal{SN}_{\mathrm{A}}(0, 1, 4)$ (**Data 7**), and $\mathcal{SN}_{\mathrm{A}}(0, 1, 6)$ (**Data 8**). Notably, the Azzalini's skew-normal noise is not

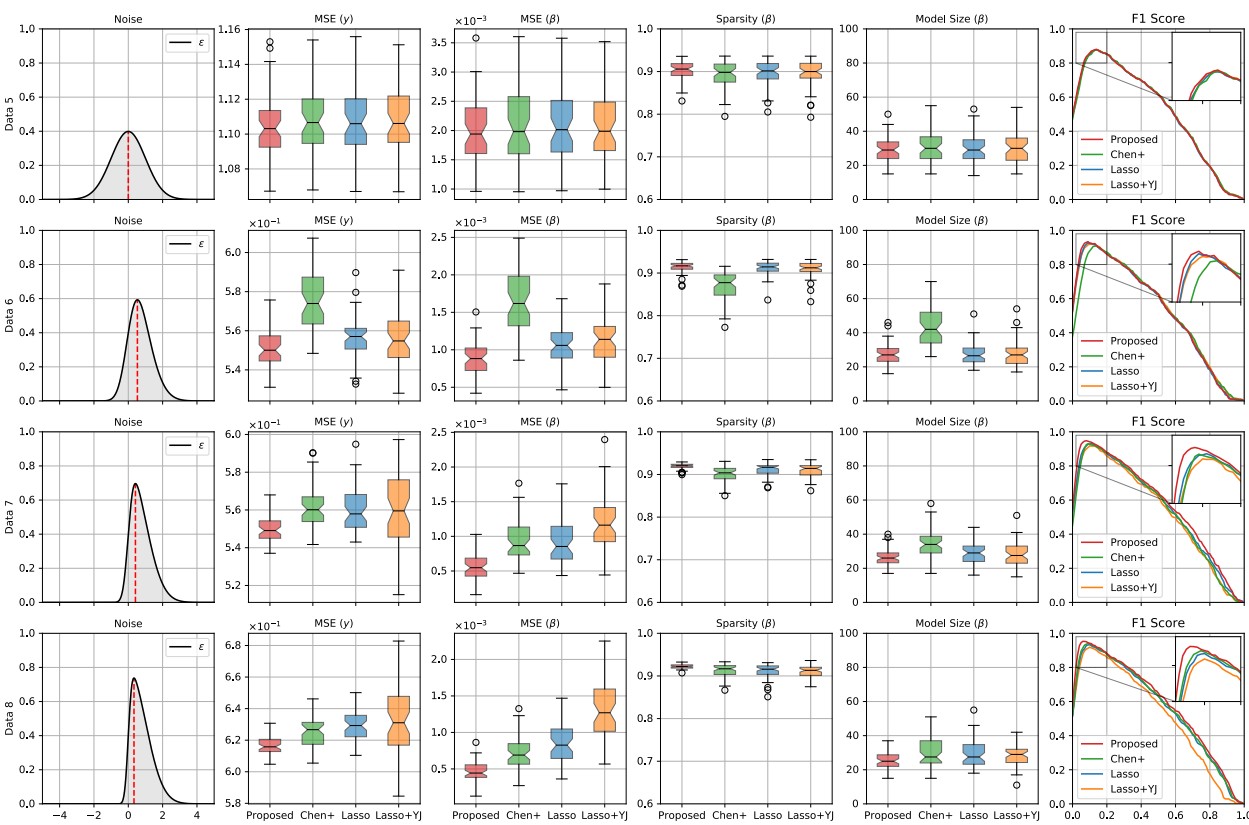

Figure 4: Comparison of four methods with four different types of simulation noises in terms of five evaluation measures. From top to bottom, each row results from Data 5, Data 6, Data 7, and Data 8, respectively. The first column plots the probability density function of the noise. The second through fifth columns are box plots of some scores. The last column is the mean F1 score for each threshold (horizontal axis). In all figures, the proposed method, Chen+ (Chen et al., 2014), Lasso, and Lasso with the Yeo-Johnson transformation are in red, green, blue, and yellow, respectively. All experiments were conducted for 50 runs with different random seeds.

mode-invariant, i.e., its location parameter does not coincide with its mode, which significantly violates the assumptions of the proposed method.

The results are shown in Figure 4. In Data 5, the four methods presented similar behaviors, as expected. In Data 6, Data 7, and Data 8, the proposed method outperformed the other methods with the difference becoming more significant as $\alpha$ increased. Surprisingly, Chen+ presented similar results to Lasso, although Chen+ was slightly worse. This would be because the objective function of Chen+ is hard to optimize due to troublesome non-convexity with complicated combinations of all parameters. Similarly, for Data 3 (skew data), Chen+ was worse than Lasso. In comparison, the proposed method can present stable behaviors because the objective function is convex for many variables $\boldsymbol{\beta} \in \mathbb{R}^P$, although it is non-convex for only two variables $\sigma, \alpha \in \mathbb{R}$, as shown in Theorem 1.

## 4.3 Real-World Medical Data

We applied the proposed method to the following two medical datasets: **PDGFR** (Platelet Derived Growth Factor Receptor) consists of $N = 79$ samples and $P = 320$ features, where the outcome is the ability to inhibit PDGFR phosphorylation (Guha & Jurs, 2004). **MTP** (MelTing Point) includes $N = 274$ samples

Table 1: Results of PDGFR ($N = 79$, $P = 320$).

| | $\mathbf{MSE}(\hat{\boldsymbol{y}})$ | $\mathbf{Sparsity}(\hat{\boldsymbol{\beta}})$ | $\mathbf{Size}(\hat{\boldsymbol{\beta}})$ |
|---|---|---|---|
| **Proposed** | $\mathbf{7.00 \times 10^{-1}}$ $(3.16 \times 10^{-1})$ | $\mathbf{9.90 \times 10^{-1}}$ $(3.16 \times 10^{-3})$ | $\mathbf{5.30 \times 10^{0}}$ $(1.84 \times 10^{0})$ |
| Chen+ | $10.9 \times 10^{-1}$ $(2.54 \times 10^{-1})$ | $8.18 \times 10^{-1}$ $(3.72 \times 10^{-1})$ | $8.17 \times 10^{0}$ $(4.82 \times 10^{0})$ |
| Lasso | $7.28 \times 10^{-1}$ $(4.79 \times 10^{-1})$ | $9.85 \times 10^{-1}$ $(9.82 \times 10^{-3})$ | $8.90 \times 10^{0}$ $(5.57 \times 10^{0})$ |
| Lasso + YJ | $12.7 \times 10^{-1}$ $(4.22 \times 10^{-1})$ | $9.85 \times 10^{-1}$ $(9.82 \times 10^{-3})$ | $8.93 \times 10^{0}$ $(5.57 \times 10^{0})$ |

Table 2: Results of MTP ($N = 274$, $P = 1142$).

| | $\mathbf{MSE}(\hat{\boldsymbol{y}})$ | $\mathbf{Sparsity}(\hat{\boldsymbol{\beta}})$ | $\mathbf{Size}(\hat{\boldsymbol{\beta}})$ |
|---|---|---|---|
| **Proposed** | $\mathbf{8.43 \times 10^{-1}}$ $(1.65 \times 10^{-1})$ | $\mathbf{9.91 \times 10^{-1}}$ $(2.05 \times 10^{-3})$ | $\mathbf{2.09 \times 10^{1}}$ $(5.45 \times 10^{0})$ |
| Chen+ | $10.8 \times 10^{-1}$ $(1.88 \times 10^{-1})$ | $9.72 \times 10^{-1}$ $(4.58 \times 10^{-3})$ | $5.93 \times 10^{1}$ $(7.47 \times 10^{0})$ |
| Lasso | $23.4 \times 10^{-1}$ $(84.9 \times 10^{-1})$ | $9.82 \times 10^{-1}$ $(6.66 \times 10^{-3})$ | $4.25 \times 10^{1}$ $(13.9 \times 10^{0})$ |
| Lasso + YJ | $11.6 \times 10^{-1}$ $(3.84 \times 10^{-1})$ | $9.82 \times 10^{-1}$ $(6.66 \times 10^{-3})$ | $4.25 \times 10^{1}$ $(13.9 \times 10^{0})$ |

and $P = 1142$ features, where the outcome is the melting point of drug-like compounds (Karthikeyan et al., 2005). These datasets have many features compared to their sample size and may hold skewed noise, as described later. We compared the proposed method with "Chen+" (Chen et al., 2014), "Lasso" (Tibshirani, 1996), and "Lasso+YJ", as in the previous section.

We used 80% of the samples for training and the remaining 20% for testing. Then, we tuned each regularization coefficient $\lambda$ with 5-fold cross-validation using 20% of the training samples (i.e., 16% of all samples) as validation data. These samples were generated randomly, and 30 trials were conducted with different random seeds. We calculated MSE($\hat{y}$), Sparsity of $\hat{\boldsymbol{\beta}}$, and Model Size($\hat{\boldsymbol{\beta}}$) and evaluated their performance.

Table 1 and Table 2 show the results of PDGFR and MTP, respectively, with the means and standard deviations (in parentheses) of the evaluation measures. For both datasets, the proposed method simultaneously achieved the smallest prediction error and the smallest model size. In particular, for MTP, the proposed method reduced the model size to less than half when compared to the other methods. Regarding the residuals of the test data after training with Lasso, the mean of the normalized skewness of the residuals was $-0.81$ for PDGFR and $0.75$ for MTP.

In Appendix I.3, we also applied the proposed method to more easily interpretable data, specifically the Engineering Graduate Salary (EGS) prediction data (Aggarwal et al., 2016). EGS has fewer features than the two datasets above, and the meanings of all features are specifically provided. These properties allow us to consider the validity of the estimated active features. The result shows that the proposed method outperformed the comparative methods and could select more reasonable features.

## 5    Conclusion

In this paper, we have proposed a novel sparse regression model whose noise is assumed to follow the mode-invariant skew-normal distribution. The proposed method always models a mode regardless of skewness and is highly adaptable and interpretable to skew noise. We have also shown that the proposed method is straightforward to implement and optimize, and it has theoretical guarantees for the average excess risk and the estimation error by non-asymptotic analysis. The results of numerical experiments demonstrated that the proposed method achieved both skewness modeling and statistical interpretability simultaneously, even for high-dimensional data, highlighting its significant effectiveness.

As a next challenge, we focus on extending to the *mode-invariant skew-t* distribution. To our knowledge, no study has explicitly dealt with this distribution besides its basic idea being mentioned in Fujisawa & Abe (2015). The first problem would be to prove the positive definiteness of the Fisher information matrix, whose proof could be complicated, such as in the proof of Proposition 2. Its optimization algorithm could also be problematic. However, these may be overcome in terms of $t$ distribution being a scale mixture of

normal distribution. It is also interesting to extend from univariate to multivariate noise for some machine learning algorithms. Fortunately, a multivariate mode-invariant skew-normal distribution has already been proposed by Abe & Fujisawa (2019), suggesting that we could quickly tackle and formulate models.

**Acknowledgments**

The authors would like to thank anonymous reviewers for their insightful comments and suggestions. This work was partially supported by JSPS KAKENHI Grant Numbers JP17K00065 and JP22K17859.

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

# A   Upper Bounds on $|H_\alpha(v)|$ and $|r_\alpha(u)|$

**Proposition 5.** *We have $|H_\alpha(v)| < \rho_\alpha|v|$, $|h_\alpha(v)| < \rho_\alpha$, $|h'_\alpha(v)| < \rho_\alpha|\alpha|$, and $|h''_\alpha(v)| < 3\rho_\alpha\alpha^2$. From $\rho_\alpha < 1$, we also have $|H_\alpha(v)| < |v|$, $|h_\alpha(v)| < 1$, $|h'_\alpha(v)| < |\alpha|$, and $|h''_\alpha(v)| < 3\alpha^2$.*

*Proof.* It holds after simple calculation that

$$h_\alpha(v) = \frac{d}{dv}H_\alpha(v) = \frac{\rho_\alpha\alpha v}{\sqrt{1+\alpha^2v^2}}, \tag{29}$$

$$h'_\alpha(v) = \frac{d^2}{dv^2}H_\alpha(v) = \frac{\rho_\alpha\alpha}{(1+\alpha^2v^2)^{\frac{3}{2}}}, \tag{30}$$

$$h''_\alpha(v) = \frac{d^3}{dv^3}H_\alpha(v) = -\frac{3\rho_\alpha\alpha^3v}{(1+\alpha^2v^2)^{\frac{5}{2}}}. \tag{31}$$

We clearly see $|h_\alpha(v)| < \rho_\alpha$ and $|h'_\alpha(v)| < \rho_\alpha|\alpha|$. Since $H_\alpha(v) = \int_0^v h_\alpha(v)dv$, we have $|H_\alpha(v)| < \rho_\alpha|v|$. We also have

$$|h''_\alpha(v)| = 3\rho_\alpha\alpha^2 \frac{|\alpha v|}{(1+\alpha^2v^2)^{\frac{1}{2}}} \frac{1}{(1+\alpha^2v^2)^2} \le 3\rho_\alpha\alpha^2 \tag{32}$$

$\square$

**Proposition 6.** *Suppose that Assumption 1 holds. Then, $r'_\alpha(u)$, $r''_\alpha(u)$, and $r'''_\alpha(u)$ are bounded and $|r_\alpha(u)|$ are bounded by $c|u|$ with some constant $c$.*

*Proof.* For $\alpha = 0$, since $r_\alpha(u) = u$, this proposition clearly holds. Hereafter, we consider the case $\alpha \neq 0$. By differentiating $u = q_\alpha(r(u))$ with respect to $u$, we have $1 = q'_\alpha(r_\alpha(u))r'_\alpha(u)$ and then

$$r'_\alpha(u) = \frac{1}{q'_\alpha(r_\alpha(u))} = \frac{1}{1+h_\alpha(r_\alpha(u))}. \tag{33}$$

From Proposition 5, we have

$$0 < \frac{1}{1+\rho_\alpha} \le r'_\alpha(u) \le \frac{1}{1-\rho_\alpha}. \tag{34}$$

We can easily see that $1/2 < \rho_\alpha < c$ with some constant $c < 1$ because $\rho_\alpha = 1 - (1/2) \exp(-\alpha^2)$ and $\alpha$ is bounded from Assumption 1. This implies that $r'_\alpha(u)$ is bounded. From $r_\alpha(0) = 0$, we have $r_\alpha(u) = \int_0^u r'_\alpha(s)ds$ and then

$$|r_\alpha(u)| \leq \frac{1}{1 - \rho_\alpha}|u| < c|u|. \tag{35}$$

By differentiating (33) with respect to $u$, we have

$$r''_\alpha(u) = -\frac{h'_\alpha(r_\alpha(u))r'_\alpha(u)}{\{1 + h_\alpha(r_\alpha(u))\}^2} = -h'_\alpha(r_\alpha(u))\{r'_\alpha(u)\}^3. \tag{36}$$

By differentiating the above with respect to $u$, we have

$$r'''_\alpha(u) = -h''_\alpha(r_\alpha(u))\{r'_\alpha(u)\}^4 - 3h'_\alpha(r_\alpha(u))\{r'_\alpha(u)\}^2 r''_\alpha(u). \tag{37}$$

From Proposition 5 and the boundedness of $\alpha$, we can easily see $r''_\alpha(u)$ and $r'''_\alpha(u)$ are bounded. $\qquad\square$

**Proposition 7.** *Fix $\alpha \neq 0$. For the definition in (4), the upper bounds of $\left|\frac{\partial}{\partial\alpha}H_\alpha(v)\right|$, $\left|\frac{\partial^2}{\partial\alpha^2}H_\alpha(v)\right|$, and $\left|\frac{\partial^3}{\partial\alpha^3}H_\alpha(v)\right|$ can be described by a linear function of $|v|$. They can also be described by a linear function of $|u|$. For $\alpha = 0$, the upper bounds of $\left|\frac{\partial}{\partial\alpha}H_\alpha(v)\right|$, $\left|\frac{\partial^2}{\partial\alpha^2}H_\alpha(v)\right|$, and $\left|\frac{\partial^3}{\partial\alpha^3}H_\alpha(v)\right|$ can be described by a quadratic function of $|v|$, a constant, and a quartic function of $|v|$, respectively. They can also be described by a quadratic function of $|u|$, a constant, and a quartic function of $|u|$, respectively.*

*Proof.* Analytically, for fixed $\alpha \neq 0$,

$$\frac{\partial}{\partial\alpha}H_\alpha(v) = \left(\frac{\alpha \exp(-\alpha^2)}{\rho_\alpha} + \frac{1}{\alpha\sqrt{1 + \alpha^2 v^2}}\right) H_\alpha(v)$$
$$\leq \left(\frac{\alpha \exp(-\alpha^2)}{\rho_\alpha} + \frac{1}{\alpha}\right) H_\alpha(v), \tag{38}$$

$$\frac{\partial^2}{\partial\alpha^2}H_\alpha(v) = \left(\frac{(\rho_\alpha - 2\alpha^2)\exp(-\alpha^2)}{\rho_\alpha^2} - \frac{v^2}{(1 + \alpha^2 v^2)^{\frac{3}{2}}} - \frac{1}{\alpha^2\sqrt{1 + \alpha^2 v^2}}\right) H_\alpha(v)$$
$$+ \left(\frac{\alpha \exp(-\alpha^2)}{\rho_\alpha} + \frac{1}{\alpha\sqrt{1 + \alpha^2 v^2}}\right)^2 H_\alpha(v)$$
$$\leq \left(\frac{|\rho_\alpha - 2\alpha^2|\exp(-\alpha^2)}{\rho_\alpha^2} + \frac{2\sqrt{3}}{9\alpha^2} + \frac{1}{\alpha^2}\right) H_\alpha(v)$$
$$+ \left(\frac{\alpha \exp(-\alpha^2)}{\rho_\alpha} + \frac{1}{\alpha}\right)^2 H_\alpha(v), \tag{39}$$

$$\frac{\partial^3}{\partial\alpha^3}H_\alpha(v) = \left(-\frac{2\alpha(\rho_\alpha - 2\alpha)\exp(-\alpha^2)}{\rho_\alpha^2} - \frac{2\alpha(\rho_\alpha - 2\alpha^2)\exp(-2\alpha^2)}{\rho_\alpha^3} - \frac{2\alpha(1+\rho_\alpha)\exp(-\alpha^2)}{\rho_\alpha^2}\right)H_\alpha(v)$$

$$+ \left(\frac{3\alpha v^4}{(1+\alpha^2 v^2)^{\frac{5}{2}}} + \frac{v^2}{\alpha(1+\alpha^2 v^2)^{\frac{3}{2}}} + \frac{2}{\alpha^3\sqrt{1+\alpha^2 v^2}}\right)H_\alpha(v)$$

$$+ 3\left(\frac{(\rho_\alpha - 2\alpha^2)\exp(-\alpha^2)}{\rho_\alpha^2} - \frac{v^2}{(1+\alpha^2 v^2)^{\frac{3}{2}}} - \frac{1}{\alpha^2\sqrt{1+\alpha^2 v^2}}\right)\left(\frac{\alpha\exp(-\alpha^2)}{\rho_\alpha} + \frac{1}{\alpha\sqrt{1+\alpha^2 v^2}}\right)H_\alpha(v)$$

$$+ \left(\frac{\alpha\exp(-\alpha^2)}{\rho_\alpha} + \frac{1}{\alpha\sqrt{1+\alpha^2 v^2}}\right)^3 H_\alpha(v)$$

$$\le \left(\frac{2\alpha|\rho_\alpha - 2\alpha|\exp(-\alpha^2)}{\rho_\alpha^2} + \frac{2\alpha|\rho_\alpha - 2\alpha^2|\exp(-2\alpha^2)}{\rho_\alpha^3} + \frac{2\alpha(1+\rho_\alpha)\exp(-\alpha^2)}{\rho_\alpha^2}\right)H_\alpha(v)$$

$$+ \frac{1}{\alpha^3}\left(\frac{48\sqrt{5}}{125} + \frac{2\sqrt{3}}{9} + 2\right)H_\alpha(v)$$

$$+ 3\left(\frac{|\rho_\alpha - 2\alpha^2|\exp(-2\alpha^2)}{\rho_\alpha^2} + \frac{2\sqrt{3}}{9\alpha^2} + \frac{1}{\alpha^2}\right)\left(\frac{\alpha\exp(-2\alpha^2)}{\rho_\alpha} + \frac{1}{\alpha}\right)H_\alpha(v)$$

$$+ \left(\frac{\alpha\exp(-\alpha^2)}{\rho_\alpha} + \frac{1}{\alpha}\right)^3 H_\alpha(v). \tag{40}$$

Therefore, since $|H_\alpha(v)| < \rho_\alpha|v|$ from Proposition 5, it holds that the upper bounds of $\left|\frac{\partial}{\partial\alpha}H_\alpha(v)\right|$, $\left|\frac{\partial^2}{\partial\alpha^2}H_\alpha(v)\right|$, and $\left|\frac{\partial^3}{\partial\alpha^3}H_\alpha(v)\right|$ can be described by a linear function of $|v|$. Furthermore, from (35), their upper bounds can also be described by a linear function of $|u|$.

Next, we consider the case $\alpha \to 0$. Let $g(x) := \sqrt{1+x^2} - 1$. Letting the $m$-th order derivative of $g(x)$ be denoted by $g^{(m)}(x)$, we have

$$g^{(1)}(x) = \frac{x}{\sqrt{1+x^2}}, \tag{41}$$

$$g^{(2)}(x) = \frac{1}{(1+x^2)^{\frac{3}{2}}}, \tag{42}$$

$$g^{(3)}(x) = -\frac{3x}{(1+x^2)^{\frac{5}{2}}}, \tag{43}$$

Then, it holds after simple calculations that

$$\lim_{\alpha\to 0}\frac{g(\alpha v)}{\alpha} = 0 \tag{44}$$

$$\lim_{\alpha\to 0}\frac{\partial}{\partial\alpha}\frac{g(\alpha v)}{\alpha} = \lim_{\alpha\to 0}\left(-\frac{g(\alpha v)}{\alpha^2} + \frac{v\cdot g^{(1)}(\alpha v)}{\alpha}\right)$$

$$= \frac{1}{2}v^2 \tag{45}$$

$$\lim_{\alpha\to 0}\frac{\partial^2}{\partial\alpha^2}\frac{g(\alpha v)}{\alpha} = \lim_{\alpha\to 0}\left(\frac{2g(\alpha v)}{\alpha^3} - \frac{2v\cdot g^{(1)}(\alpha v)}{\alpha^2} + \frac{v^2\cdot g^{(2)}(\alpha v)}{\alpha}\right)$$

$$= 0 \tag{46}$$

$$\lim_{\alpha\to 0}\frac{\partial^3}{\partial\alpha^3}\frac{g(\alpha v)}{\alpha} = \lim_{\alpha\to 0}\left(-\frac{6g(\alpha v)}{\alpha^4} + \frac{6v\cdot g^{(1)}(\alpha v)}{\alpha^3} - \frac{3v^2\cdot g^{(2)}(\alpha v)}{\alpha^2} + \frac{v^3\cdot g^{(3)}(\alpha v)}{\alpha}\right)$$

$$= -\frac{3}{4}v^4 \tag{47}$$

It also holds that

$$\lim_{\alpha \to 0} \rho_\alpha = \lim_{\alpha \to 0} \left( 1 - \frac{1}{2} \exp(-\alpha^2) \right)$$
$$= \frac{1}{2} \tag{48}$$

$$\lim_{\alpha \to 0} \frac{\partial}{\partial \alpha} \rho_\alpha = \lim_{\alpha \to 0} \alpha \exp(-\alpha^2)$$
$$= 0 \tag{49}$$

$$\lim_{\alpha \to 0} \frac{\partial^2}{\partial \alpha^2} \rho_\alpha = \lim_{\alpha \to 0} (1 - 2\alpha^2) \exp(-\alpha^2)$$
$$= 1 \tag{50}$$

$$\lim_{\alpha \to 0} \frac{\partial^3}{\partial \alpha^3} \rho_\alpha = \lim_{\alpha \to 0} 2\alpha(2\alpha^2 - 3) \exp(-\alpha^2)$$
$$= 0 \tag{51}$$

Note that $H_\alpha(v) = \rho_\alpha \frac{g(\alpha v)}{\alpha}$. Using from (44) through (51), we have

$$\lim_{\alpha \to 0} \frac{\partial}{\partial \alpha} H_\alpha(v) = \lim_{\alpha \to 0} \left( \frac{\partial}{\partial \alpha} \rho_\alpha \cdot \frac{g(\alpha v)}{\alpha} + \rho_\alpha \cdot \frac{\partial}{\partial \alpha} \frac{g(\alpha v)}{\alpha} \right)$$
$$= \frac{1}{4} v^2 \tag{52}$$

$$\lim_{\alpha \to 0} \frac{\partial^2}{\partial \alpha^2} H_\alpha(v) = \lim_{\alpha \to 0} \left( \frac{\partial^2}{\partial \alpha^2} \rho_\alpha \cdot \frac{g(\alpha v)}{\alpha} + 2 \cdot \frac{\partial}{\partial \alpha} \rho_\alpha \cdot \frac{\partial}{\partial \alpha} \frac{g(\alpha v)}{\alpha} + \rho_\alpha \cdot \frac{\partial^2}{\partial \alpha^2} \frac{g(\alpha v)}{\alpha} \right)$$
$$= 0 \tag{53}$$

$$\lim_{\alpha \to 0} \frac{\partial^3}{\partial \alpha^3} H_\alpha(v) = \lim_{\alpha \to 0} \left( \frac{\partial^3}{\partial \alpha^3} \rho_\alpha \cdot \frac{g(\alpha v)}{\alpha} + 3 \cdot \frac{\partial^2}{\partial \alpha^2} \rho_\alpha \cdot \frac{\partial}{\partial \alpha} \frac{g(\alpha v)}{\alpha} + 3 \cdot \frac{\partial}{\partial \alpha} \rho_\alpha \cdot \frac{\partial^2}{\partial \alpha^2} \frac{g(\alpha v)}{\alpha} + \rho_\alpha \cdot \frac{\partial^3}{\partial \alpha^3} \frac{g(\alpha v)}{\alpha} \right)$$
$$= \frac{3}{2} v^2 - \frac{3}{8} v^4 \tag{54}$$

Therefore, the upper bounds of $\left| \frac{\partial}{\partial \alpha} H_\alpha(v) \right|$, $\left| \frac{\partial^2}{\partial \alpha^2} H_\alpha(v) \right|$, and $\left| \frac{\partial^3}{\partial \alpha^3} H_\alpha(v) \right|$ as $\alpha \to 0$ can be described by a quadratic function of $|v|$, a constant, and a quartic function of $|v|$, respectively. From (35), their upper bounds can also be described by a quadratic function of $|u|$, a constant, and a quartic function of $|u|$, respectively. $\square$

**Proposition 8.** *Fix $\alpha \neq 0$. For the definition in (4), the upper bounds of $\left| \frac{\partial}{\partial \alpha} r_\alpha(u) \right|$, $\left| \frac{\partial^2}{\partial \alpha^2} r_\alpha(u) \right|$, and $\left| \frac{\partial^3}{\partial \alpha^3} r_\alpha(u) \right|$ can be described by a linear function of $|u|$, a quadratic function of $|u|$, and a cubic function of $|u|$, respectively. For $\alpha = 0$, the upper bounds of $\left| \frac{\partial}{\partial \alpha} r_\alpha(u) \right|$, $\left| \frac{\partial^2}{\partial \alpha^2} r_\alpha(u) \right|$, and $\left| \frac{\partial^3}{\partial \alpha^3} r_\alpha(u) \right|$ can be described by a quadratic function of $|u|$, a quartic function of $|u|$, and a sextic function of $|u|$, respectively.*

*Proof.* Using the Euler's chain rule,

$$\frac{\partial}{\partial \alpha} r_\alpha(u) = -r'_\alpha(u) \frac{\partial}{\partial \alpha} H_\alpha(v), \tag{55}$$

$$\frac{\partial^2}{\partial \alpha^2} r_\alpha(u) = r''_\alpha(u) \left( \frac{\partial}{\partial \alpha} H_\alpha(v) \right)^2 - r'_\alpha(u) \frac{\partial^2}{\partial \alpha^2} H_\alpha(v), \tag{56}$$

$$\frac{\partial^3}{\partial \alpha^3} r_\alpha(u) = -r'''_\alpha(u) \left( \frac{\partial}{\partial \alpha} H_\alpha(v) \right)^3 + 3r''_\alpha(u) \frac{\partial}{\partial \alpha} H_\alpha(v) \cdot \frac{\partial^2}{\partial \alpha^2} H_\alpha(v) - r'_\alpha(u) \frac{\partial^3}{\partial \alpha^3} H_\alpha(v). \tag{57}$$

Therefore, from Proposition 6 and Proposition 7, the proof is complete. $\square$

## B Proof of Theorem 1

Let $f_*(u \mid \alpha) := \phi(r_\alpha(u))$. Let $l_*(\alpha \mid u) := \log f_*(u \mid \alpha)$. For $\alpha = 0$, since the optimization problem can be regarded as an ordinary least squares method, this theorem clearly holds. Therefore, in the following, we consider the case $\alpha \neq 0$. From $u = (y - \mu)/\sigma$ and $\mu = \boldsymbol{X}^\top \boldsymbol{\beta}$, since

$$
\frac{\partial^2}{\partial \boldsymbol{\beta} \partial \boldsymbol{\beta}^\top} \ell(\boldsymbol{\theta} \mid \mathcal{D}_N) = -\frac{1}{N} \sum_{n=1}^N \frac{\partial^2}{\partial \boldsymbol{\beta} \partial \boldsymbol{\beta}^\top} l(\boldsymbol{\beta}, \sigma, \alpha \mid \boldsymbol{X}_n, y_n)
$$

$$
= -\frac{1}{N\sigma^2} \sum_{n=1}^N \boldsymbol{X}_n \boldsymbol{X}_n^\top \frac{\partial^2}{\partial u_n^2} l_*(\alpha \mid u_n), \tag{58}
$$

it is sufficient for Theorem 1 to verify the convexity of $l_*(\alpha \mid u_n)$ for $u_n$. Letting $v_n := r_\alpha(u_n)$, we have

$$
-\frac{\partial^2}{\partial u_n^2} l_*(\alpha \mid u_n) = \frac{1}{2} \frac{\partial^2}{\partial u_n^2} r_\alpha^2(u_n)
$$

$$
= \left( r_\alpha'(u_n) \right)^2 + r_\alpha(u_n) r_\alpha''(u_n)
$$

$$
= \frac{1 + h_\alpha(v_n) - v_n h_\alpha'(v_n)}{\left( 1 + h_\alpha(v_n) \right)^3}, \tag{59}
$$

where the last equality holds from the calculation in the proof of Proposition 6.

Since $|h_\alpha(v_n)| < \rho_\alpha < 1$ from Proposition 5, the denominator of (59) is bounded by

$$
0 < (1 - \rho_\alpha)^3 < (1 + h_\alpha(v_n))^3 < (1 + \rho_\alpha)^3 < 8. \tag{60}
$$

The numerator of (59) is obviously 1 for $v_n = 0$. Consider the case $v_n \neq 0$. The numerator of (59) is expressed as

$$
1 + h_\alpha(v_n) - v_n h_\alpha'(v_n) = 1 - v_n^2 \left( \frac{h_\alpha(v_n)}{v_n} \right)'
$$

$$
= 1 - \rho_\alpha \left( \frac{\alpha v_n}{\sqrt{1 + \alpha^2 v_n^2}} \right)^3, \tag{61}
$$

and then

$$
0 < 1 - \rho_\alpha < 1 + h_\alpha(v_n) - v_n h_\alpha'(v_n) < 1 + \rho_\alpha < 2. \tag{62}
$$

Combining (59), (60), and (62), we have

$$
0 < \frac{1 - \rho_\alpha}{(1 + \rho_\alpha)^3} < \frac{1 + h_\alpha(v_n) - v_n h_\alpha'(v_n)}{(1 + h_\alpha(v_n))^3} < \frac{1 + \rho_\alpha}{(1 - \rho_\alpha)^3} < \infty. \tag{63}
$$

Thus, there exist some constants $C_L := (1 - \rho_\alpha)/(1 + \rho_\alpha)^3 > 0$ and $C_U := (1 + \rho_\alpha)/(1 - \rho_\alpha)^3 > 0$, satisfying

$$
\frac{C_L}{N\sigma^2} \sum_{n=1}^N \boldsymbol{X}_n \boldsymbol{X}_n^\top \prec \frac{\partial^2}{\partial \boldsymbol{\beta} \partial \boldsymbol{\beta}^\top} \ell(\boldsymbol{\theta} \mid \mathcal{D}_N) \prec \frac{C_U}{N\sigma^2} \sum_{n=1}^N \boldsymbol{X}_n \boldsymbol{X}_n^\top. \tag{64}
$$

From Assumption 1, $\alpha$ is bounded and then we see that $C_L$ is bounded above zero and $C_U$ is bounded below. In particular, if $S_N$ is positive definite, the minimum eigenvalue of $\frac{\partial^2}{\partial \boldsymbol{\beta} \partial \boldsymbol{\beta}^\top} \ell(\boldsymbol{\theta} \mid \mathcal{D}_N)$ is positive from (64). Hence, $\ell(\boldsymbol{\theta} \mid \mathcal{D}_N)$ is strongly convex for $\boldsymbol{\beta}$. The proof is complete.

## C  Derivation of MM Algorithm

Consider fixing $(\sigma, \alpha)$ and optimizing $\boldsymbol{\beta}$. According to Fujisawa & Abe (2015), the function $r_\alpha$ corresponding to (4) is

$$r_\alpha(u) = \begin{cases} \frac{1}{\alpha(1-\rho_\alpha^2)}\left(\alpha u + \rho_\alpha - \rho_\alpha\sqrt{(\alpha u + \rho_\alpha)^2 + (1-\rho_\alpha^2)}\right) & (\alpha \neq 0) \\ u & (\alpha = 0) \end{cases}. \tag{65}$$

Let $\ell_F(\boldsymbol{\beta}) := \ell(\boldsymbol{\theta} \mid \mathcal{D}_N) + \lambda\|\boldsymbol{\beta}\|_1$ with fixed $(\sigma, \alpha)$. For $\alpha \neq 0$, the optimize problem (7) is rewritten as

$$\begin{aligned}
\arg\min_{\boldsymbol{\beta}} \ell_F(\boldsymbol{\beta}) &= \arg\min_{\boldsymbol{\beta}} -\frac{1}{N}\sum_{n=1}^{N} \log f(y_n \mid \boldsymbol{X}_n^\top\boldsymbol{\beta}, \sigma, \alpha) + \lambda\|\boldsymbol{\beta}\|_1 \\
&= \arg\min_{\boldsymbol{\beta}} \frac{1}{2N}\sum_{n=1}^{N}\left(\frac{z_n - \rho_\alpha w_n}{\alpha(1-\rho_\alpha^2)}\right)^2 + \lambda\|\boldsymbol{\beta}\|_1 \\
&= \arg\min_{\boldsymbol{\beta}} \frac{1}{2N}\sum_{n=1}^{N}\frac{z_n^2 - 2\rho_\alpha z_n w_n + \rho_\alpha^2(z_n^2 + 1 - \rho_\alpha^2)}{\alpha^2(1-\rho_\alpha^2)^2} + \lambda\|\boldsymbol{\beta}\|_1 \\
&= \arg\min_{\boldsymbol{\beta}} \frac{1}{2N}\sum_{n=1}^{N}\frac{(1+\rho_\alpha^2)z_n^2 - 2\rho_\alpha z_n w_n}{\alpha^2(1-\rho_\alpha^2)^2} + \lambda\|\boldsymbol{\beta}\|_1 \\
&= \arg\min_{\boldsymbol{\beta}} \frac{1+\rho_\alpha^2}{2N\alpha^2(1-\rho_\alpha^2)^2}\sum_{n=1}^{N}\left(z_n^2 - \frac{2\rho_\alpha}{1+\rho_\alpha^2}z_n w_n\right) + \lambda\|\boldsymbol{\beta}\|_1,
\end{aligned} \tag{66}$$

where $z_n(\boldsymbol{\beta}) := \frac{\alpha}{\sigma}(y_n - \boldsymbol{X}_n^\top\boldsymbol{\beta}) + \rho_\alpha$ and $w_n(\boldsymbol{\beta}) := \sqrt{z_n^2 + 1 - \rho_\alpha^2}$. Thus, there exists some constant $C_F$, satisfying

$$\ell_F(\boldsymbol{\beta}) = \frac{1+\rho_\alpha^2}{2N\alpha^2(1-\rho_\alpha^2)^2}\sum_{n=1}^{N}\left(z_n^2 - \frac{2\rho_\alpha}{1+\rho_\alpha^2}z_n w_n\right) + \lambda\|\boldsymbol{\beta}\|_1 + C_F. \tag{67}$$

Note that $\frac{\partial}{\partial\boldsymbol{\beta}}z_n = -\frac{\alpha}{\sigma}\boldsymbol{X}_n$ and $\frac{\partial}{\partial\boldsymbol{\beta}}w_n = -\frac{\alpha}{\sigma}\frac{z_n}{w_n}\boldsymbol{X}_n$. Letting $\hat{l}_n(\boldsymbol{\beta}) := z_n^2 - \frac{2\rho_\alpha}{1+\rho_\alpha^2}z_n w_n$, we have

$$\begin{aligned}
\frac{\partial}{\partial\boldsymbol{\beta}}\hat{l}_n(\boldsymbol{\beta}) &= -\frac{2\alpha}{\sigma}z_n\boldsymbol{X}_n + \frac{2\alpha\rho_\alpha}{\sigma(1+\rho_\alpha^2)}\left(w_n + \frac{z_n^2}{w_n}\right)\boldsymbol{X}_n \\
&= -\frac{2\alpha}{\sigma}z_n\boldsymbol{X}_n + \frac{2\alpha\rho_\alpha}{\sigma(1+\rho_\alpha^2)}\left(2w_n - \frac{1-\rho_\alpha^2}{w_n}\right)\boldsymbol{X}_n
\end{aligned} \tag{68}$$

$$\frac{\partial^2}{\partial\boldsymbol{\beta}\partial\boldsymbol{\beta}^\top}\hat{l}_n(\boldsymbol{\beta}) = \frac{2\alpha^2}{\sigma^2}\boldsymbol{X}_n\boldsymbol{X}_n^\top - \frac{2\alpha^2\rho_\alpha}{\sigma^2(1+\rho_\alpha^2)}\left(2 + \frac{1-\rho_\alpha^2}{w_n^2}\right)\frac{z_n}{w_n}\boldsymbol{X}_n\boldsymbol{X}_n^\top. \tag{69}$$

Since $\left(2 + \frac{1-\rho_\alpha^2}{w_n^2}\right)\frac{z_n}{w_n} \to \pm 2$ as $z_n \to \pm\infty$ and

$$\frac{\partial}{\partial z_n}\left(2 + \frac{1-\rho_\alpha^2}{w_n^2}\right)\frac{z_n}{w_n} = \frac{3(1-\rho_\alpha^2)^2}{w_n^5} > 0, \tag{70}$$

we have $\left|\left(2 + \frac{1-\rho_\alpha^2}{w_n^2}\right)\frac{z_n}{w_n}\right| < 2$. Then, using $0 \leq \frac{\rho_\alpha}{1+\rho_\alpha^2} < \frac{1}{2}$ from $0 \leq \rho_\alpha < 1$, we have from (69),

$$\boldsymbol{O} \prec \frac{\partial^2}{\partial\boldsymbol{\beta}\partial\boldsymbol{\beta}^\top}\hat{l}_n(\boldsymbol{\beta}) \prec \frac{4\alpha^2}{\sigma^2}\boldsymbol{X}_n\boldsymbol{X}_n^\top. \tag{71}$$

From (67) and (71), we can construct a surrogate function $\ell_G(\boldsymbol{\beta} \mid \boldsymbol{\beta}^{(t)})$ of $\ell_F(\boldsymbol{\beta})$ near $\boldsymbol{\beta}^{(t)}$ based on the second-order Taylor expansion as follows (see Lemma 12 of Sun et al. (2016) for details):

$$
\begin{aligned}
\ell_F(\boldsymbol{\beta}) \leq & \frac{1+\rho_\alpha^2}{2N\alpha^2(1-\rho_\alpha^2)^2} \sum_{n=1}^{N} \left( \hat{l}_n(\boldsymbol{\beta}^{(t)}) + \left.\frac{\partial \hat{l}_n}{\partial \boldsymbol{\beta}}\right|_{\boldsymbol{\beta}=\boldsymbol{\beta}^{(t)}} (\boldsymbol{\beta} - \boldsymbol{\beta}^{(t)}) + \frac{2\alpha^2}{\sigma^2} (\boldsymbol{\beta} - \boldsymbol{\beta}^{(t)})^\top \boldsymbol{X}_n \boldsymbol{X}_n^\top (\boldsymbol{\beta} - \boldsymbol{\beta}^{(t)}) \right) \\
& + \lambda \|\boldsymbol{\beta}\|_1 + C_F \\
=: & \ell_G(\boldsymbol{\beta} \mid \boldsymbol{\beta}^{(t)}).
\end{aligned}
\tag{72}
$$

Therefore, from (68) and (72), we can update $\boldsymbol{\beta}^{(t)}$ at the $t$-th iteration as follows:

$$
\begin{aligned}
\boldsymbol{\beta}^{(t+1)} &= \arg\min_{\boldsymbol{\beta}} \ell_G(\boldsymbol{\beta} \mid \boldsymbol{\beta}^{(t)}) \\
&= \arg\min_{\boldsymbol{\beta}} \frac{1+\rho_\alpha^2}{2N\alpha^2(1-\rho_\alpha^2)^2} \sum_{n=1}^{N} \left( \left.\frac{\partial \hat{l}_n}{\partial \boldsymbol{\beta}}\right|_{\boldsymbol{\beta}=\boldsymbol{\beta}^{(t)}} (\boldsymbol{\beta} - \boldsymbol{\beta}^{(t)}) + \frac{2\alpha^2}{\sigma^2} (\boldsymbol{\beta} - \boldsymbol{\beta}^{(t)})^\top \boldsymbol{X}_n \boldsymbol{X}_n^\top (\boldsymbol{\beta} - \boldsymbol{\beta}^{(t)}) \right) + \lambda \|\boldsymbol{\beta}\|_1 \\
&= \arg\min_{\boldsymbol{\beta}} \frac{1+\rho_\alpha^2}{N\sigma^2(1-\rho_\alpha^2)^2} \sum_{n=1}^{N} \left( \tilde{y}_n - \boldsymbol{X}_n^\top \boldsymbol{\beta} \right)^2 + \lambda \|\boldsymbol{\beta}\|_1,
\end{aligned}
\tag{73}
$$

where $\tilde{y}_n := \boldsymbol{X}_n^\top \boldsymbol{\beta}^{(t)} + \frac{\sigma}{2\alpha} \left( z_n^{(t)} - \frac{\rho_\alpha}{1+\rho_\alpha^2} \left( w_n^{(t)} + \frac{z_n^{(t)2}}{w_n^{(t)}} \right) \right)$ with $z_n^{(t)} := z_n(\boldsymbol{\beta}^{(t)})$ and $w_n^{(t)} := w_n(\boldsymbol{\beta}^{(t)})$.

For $\alpha = 0$, the optimize problem (7) is rewritten as the ordinary Lasso, i.e.,

$$
\arg\min_{\boldsymbol{\beta}} \frac{1}{2N\sigma^2} \sum_{n=1}^{N} \left( y_n - \boldsymbol{X}_n^\top \boldsymbol{\beta} \right)^2 + \lambda \|\boldsymbol{\beta}\|_1.
\tag{74}
$$

## D   Proof of Theorem 2

First, we fix $(\sigma, \alpha)$ and consider the convergence of the MM algorithm for $\boldsymbol{\beta}$. We will prove that the surrogate function $\ell_G(\boldsymbol{\beta} \mid \boldsymbol{\beta}^{(t)})$ in Appendix C is a first-order surrogate of $\ell_F(\boldsymbol{\beta})$ near $\boldsymbol{\beta}^{(t)}$ based on Definition 2.1 of Mairal (2013).

From (72), $\ell_G(\boldsymbol{\beta} \mid \boldsymbol{\beta}^{(t)})$ is obviously a majorant function of $\ell_F(\boldsymbol{\beta})$. From (64) and (71), we have

$$
\frac{\partial^2}{\partial \boldsymbol{\beta} \partial \boldsymbol{\beta}^\top} \ell_F(\boldsymbol{\beta}) \prec \frac{C_U}{N\sigma^2} \sum_{n=1}^{N} \boldsymbol{X}_n \boldsymbol{X}_n^\top \prec \underbrace{\frac{C_U}{\sigma^2} \Lambda_{\max}(S_N)}_{=:L_F} \boldsymbol{I},
\tag{75}
$$

$$
\frac{\partial^2}{\partial \boldsymbol{\beta} \partial \boldsymbol{\beta}^\top} \ell_G(\boldsymbol{\beta} \mid \boldsymbol{\beta}^{(t)}) \prec \frac{2(1+\rho_\alpha^2)}{N\sigma^2(1-\rho_\alpha^2)^2} \sum_{n=1}^{N} \boldsymbol{X}_n \boldsymbol{X}_n^\top \prec \underbrace{\frac{2(1+\rho_\alpha^2)}{\sigma^2(1-\rho_\alpha^2)^2} \Lambda_{\max}(S_N)}_{=:L_G} \boldsymbol{I},
\tag{76}
$$

where $\Lambda_{\max}(S_N)$ is the largest eigenvalue of $S_N := \frac{1}{N} \sum_{n=1}^{N} \boldsymbol{X}_n \boldsymbol{X}_n^\top$. Hence, $\frac{\partial}{\partial \boldsymbol{\beta}} \ell_F(\boldsymbol{\beta})$ and $\frac{\partial}{\partial \boldsymbol{\beta}} \ell_G(\boldsymbol{\beta} \mid \boldsymbol{\beta}^{(t)})$ are $L_F$-Lipschitz continuous and $L_G$-Lipschitz continuous, respectively. Let $\ell_H(\boldsymbol{\beta} \mid \boldsymbol{\beta}^{(t)}) := \ell_G(\boldsymbol{\beta} \mid \boldsymbol{\beta}^{(t)}) - \ell_F(\boldsymbol{\beta})$. Then, $\frac{\partial}{\partial \boldsymbol{\beta}} \ell_H(\boldsymbol{\beta} \mid \boldsymbol{\beta}^{(t)})$ is also $(L_F + L_G)$-Lipschitz continuous because for any $\boldsymbol{\beta}_{\mathbf{x}}$ and $\boldsymbol{\beta}_{\mathbf{y}}$,

$$
\begin{aligned}
\left\| \frac{\partial}{\partial \boldsymbol{\beta}} \ell_H(\boldsymbol{\beta}_{\mathbf{x}} \mid \boldsymbol{\beta}^{(t)}) - \frac{\partial}{\partial \boldsymbol{\beta}} \ell_H(\boldsymbol{\beta}_{\mathbf{y}} \mid \boldsymbol{\beta}^{(t)}) \right\| &\leq \left\| \frac{\partial}{\partial \boldsymbol{\beta}} \ell_G(\boldsymbol{\beta}_{\mathbf{x}} \mid \boldsymbol{\beta}^{(t)}) - \frac{\partial}{\partial \boldsymbol{\beta}} \ell_G(\boldsymbol{\beta}_{\mathbf{y}} \mid \boldsymbol{\beta}^{(t)}) \right\| + \left\| \frac{\partial}{\partial \boldsymbol{\beta}} \ell_F(\boldsymbol{\beta}_{\mathbf{x}}) - \frac{\partial}{\partial \boldsymbol{\beta}} \ell_F(\boldsymbol{\beta}_{\mathbf{y}}) \right\| \\
&\leq L_G \|\boldsymbol{\beta}_{\mathbf{x}} - \boldsymbol{\beta}_{\mathbf{y}}\| + L_F \|\boldsymbol{\beta}_{\mathbf{x}} - \boldsymbol{\beta}_{\mathbf{y}}\| \\
&= (L_F + L_G) \|\boldsymbol{\beta}_{\mathbf{x}} - \boldsymbol{\beta}_{\mathbf{y}}\|.
\end{aligned}
\tag{77}
$$

Moreover, we have $\ell_H(\boldsymbol{\beta}^{(t)} \mid \boldsymbol{\beta}^{(t)}) = 0$ and $\frac{\partial}{\partial \boldsymbol{\beta}} \ell_H(\boldsymbol{\beta}^{(t)} \mid \boldsymbol{\beta}^{(t)}) = 0$.

Therefore, $\ell_G(\boldsymbol{\beta} \mid \boldsymbol{\beta}^{(t)})$ is a first-order surrogate of $\ell_F(\boldsymbol{\beta})$ near $\boldsymbol{\beta}^{(t)}$. From Proposition 2.2 of Mairal (2013), we have for any $t \geq 1$ and some positive constant $R$,

$$\tilde{\ell}(\boldsymbol{\beta}^{(t)}, \sigma, \alpha) - \tilde{\ell}(\boldsymbol{\beta}^*, \sigma, \alpha) \leq \frac{2(L_F + L_G)R^2}{t + 2}, \tag{78}$$

where $\tilde{\ell}(\boldsymbol{\theta}) := \ell(\boldsymbol{\theta} \mid \mathcal{D}_N) + \lambda\|\boldsymbol{\beta}\|_1$ and $\boldsymbol{\beta}^*$ is a minimizer of $\tilde{\ell}$ with fixed $(\sigma, \alpha)$. Hence, it holds that $\boldsymbol{\beta}^{(t)} \to \boldsymbol{\beta}^*$ as $t \to \infty$.

Next, we regard Algorithm 1 as the block coordinate descent (BCD) method for the three blocks $(\boldsymbol{\beta}, \sigma, \alpha)$ and consider its convergence based on Theorem 4.1 (c) of Tseng (2001). When using the cyclic rule (see Section 2 of Tseng (2001)) to update the BCD method for $B$ blocks, Theorem 4.1 (c) of Tseng (2001) states the following.

**Theorem 4.1 (c) of Tseng (2001).** *Assume that the level set $\boldsymbol{\Theta}^0 := \left\{\boldsymbol{\theta} \mid \tilde{\ell}(\boldsymbol{\theta} \mid \mathcal{D}_N) \leq \tilde{\ell}(\boldsymbol{\theta}^{(0)} \mid \mathcal{D}_N)\right\}$ is compact and that $\tilde{\ell}$ is continuous on $\boldsymbol{\Theta}^0$. If $\tilde{\ell}(\boldsymbol{\theta}_1, \ldots, \boldsymbol{\theta}_B)$ has at most one minimum in $\boldsymbol{\theta}_b$ for $b = 2, \ldots, B-1$, then every cluster point $\boldsymbol{\theta}^\dagger$ of $\{\boldsymbol{\theta}^{(t)}\}_{t := (B-1) \bmod B}$ is a coordinatewise minimum point of $\tilde{\ell}$. In addition, if $\tilde{\ell}$ is regular at $\boldsymbol{\theta}^\dagger$, then $\boldsymbol{\theta}^\dagger$ is a stationary point of $\tilde{\ell}$.*

We assume that there always exists $(y_n, \boldsymbol{X}_n) \in \mathcal{D}_N$ such that $y_n - \boldsymbol{X}_n^\top \boldsymbol{\beta} > 0$ after updating $\boldsymbol{\beta}$ by the MM algorithm, and also, there always exists $(y_n, \boldsymbol{X}_n) \in \mathcal{D}_N$ such that $y_n - \boldsymbol{X}_n^\top \boldsymbol{\beta} < 0$. Under this assumption, it holds that $\tilde{\ell}(\boldsymbol{\theta} \mid \mathcal{D}_N) \to \infty$ as $\sigma \downarrow 0$, $\sigma \to \infty$, $\alpha \to -\infty$, or $\alpha \to \infty$.

Let $\mathrm{dom}\,\ell$ and $\mathrm{dom}\,\tilde{\ell}$ be the effective domain of $\ell$ and $\tilde{\ell}$, respectively. We can see that $\tilde{\ell}(\boldsymbol{\theta} \mid \mathcal{D}_N)$ is continuous in $\mathrm{dom}\,\tilde{\ell}$ and the level set $\boldsymbol{\Theta}^0$ is compact. From Lemma 3.1 of Tseng (2001), $\tilde{\ell}$ is regular at any $\boldsymbol{\theta} \in \mathrm{dom}\,\tilde{\ell}$ since $\mathrm{dom}\,\ell$ is open and $\ell$ is Gâteaux-differentiable on $\mathrm{dom}\,\ell$. Therefore, from Theorem 4.1 (c) of Tseng (2001), if $\tilde{\ell}(\boldsymbol{\theta} \mid \mathcal{D}_N)$ has at most one minimum in $\sigma$, then Theorem 2 holds. We will prove that this necessary condition holds.

Let $\chi(u) := u r_\alpha(u) r'_\alpha(u)$. We have

$$\begin{aligned} \chi'(u) &= r_\alpha(u) r'_\alpha(u) + u\left(r'_\alpha(u)\right)^2 + u r_\alpha(u) r''_\alpha(u) \\ &= r_\alpha(u) r'_\alpha(u) + u\left(\left(r'_\alpha(u)\right)^2 + r_\alpha(u) r''_\alpha(u)\right). \end{aligned} \tag{79}$$

Since $r'_\alpha(u) > 0$ from (34) and $r_\alpha(0) = 0$ from (65), it holds that

$$\begin{aligned} \mathrm{sgn}(r_\alpha(u) r'_\alpha(u)) &= \mathrm{sgn}(r(u)) \\ &= \mathrm{sgn}(u). \end{aligned} \tag{80}$$

Since $\left(r'_\alpha(u)\right)^2 + r_\alpha(u) r''_\alpha(u) > 0$ from (59) and (63), it holds that

$$\mathrm{sgn}\left(u\left(\left(r'_\alpha(u)\right)^2 + r_\alpha(u) r''_\alpha(u)\right)\right) = \mathrm{sgn}(u). \tag{81}$$

Hence, we have

$$\mathrm{sgn}(\chi'(u)) = \mathrm{sgn}(u). \tag{82}$$

Let $u_n := \frac{y_n - \boldsymbol{X}_n^\top \boldsymbol{\beta}}{\sigma}$. For $y_n - \boldsymbol{X}_n^\top \boldsymbol{\beta} > 0$, $u_n \in (0, \infty)$ decreases monotonically as $\sigma$ increases. Since $\chi'(u_n) > 0$ for $u_n \in (0, \infty)$ from (82), $\chi(u_n)$ is a strictly decreasing function for $\sigma$. For $y_n - \boldsymbol{X}_n^\top \boldsymbol{\beta} < 0$, $u_n \in (-\infty, 0)$ increases monotonically as $\sigma$ increases. Since $\chi'(u_n) < 0$ for $u_n \in (-\infty, 0)$ from (82), $\chi(u_n)$ is also a strictly decreasing function for $\sigma$. For $y_n - \boldsymbol{X}_n^\top \boldsymbol{\beta} = 0$, $\chi(u_n) = \chi(0) = 0$.

Thus, $\frac{\partial}{\partial\sigma}\ell(\boldsymbol{\theta} \mid \mathcal{D}_N)$ is a strictly increasing function for $\sigma$, because

$$
\begin{aligned}
\frac{\partial}{\partial\sigma}\ell(\boldsymbol{\theta} \mid \mathcal{D}_N) &= -\frac{1}{N}\sum_{n=1}^{N}\frac{\partial}{\partial\sigma}l(\boldsymbol{\theta} \mid \boldsymbol{X}_n, y_n) \\
&= \frac{1}{N\sigma}\sum_{n=1}^{N}(1 - u_n r_\alpha(u_n) r'_\alpha(u_n)) \\
&= \frac{1}{N\sigma}\left(N - \sum_{n=1}^{N}\chi(u_n)\right).
\end{aligned}
\tag{83}
$$

Moreover, there exists only one $\sigma(=: \tilde{\sigma})$ satisfying $\sum_{n=1}^{N}\chi(u_n) = N$, because

$$
\begin{aligned}
\lim_{\sigma\downarrow 0}\sum_{n=1}^{N}\chi(u_n) &= \underbrace{\lim_{u_n\to-\infty}\sum_{u_n<0}\chi(u_n)}_{\to\infty} + \underbrace{\lim_{u_n\to\infty}\sum_{u_n>0}\chi(u_n)}_{\to\infty} \\
&= \infty,
\end{aligned}
\tag{84}
$$

$$
\begin{aligned}
\lim_{\sigma\to\infty}\sum_{n=1}^{N}\chi(u_n) &= \underbrace{\lim_{u_n\uparrow 0}\sum_{u_n<0}\chi(u_n)}_{\to 0} + \underbrace{\lim_{u_n\downarrow 0}\sum_{u_n>0}\chi(u_n)}_{\to 0} \\
&= 0.
\end{aligned}
\tag{85}
$$

$\tilde{\sigma}$ is also the only solution satisfying $\frac{\partial}{\partial\sigma}\ell(\boldsymbol{\theta} \mid \mathcal{D}_N) = 0$ and the only minimum of $\ell(\boldsymbol{\theta} \mid \mathcal{D}_N)$. The proof is complete.

Note that from Theorem 1, if $S_N$ is positive definite, $\ell(\boldsymbol{\theta} \mid \mathcal{D}_N)$ is strongly convex for $\boldsymbol{\beta}$ and has only one minimum in $\boldsymbol{\beta}$. In this case, $\sigma$ can be optimized first by swapping lines 2 to 9 and line 10 in Algorithm 1, and then every cluster point in a sequence of $\boldsymbol{\theta}$ after updating $\boldsymbol{\beta}$ is a stationary point.

## E  Proof of Propositions

### E.1  Preliminary Propositions

**Proposition 9.** *The $M$-th order moment of $U \sim f_*(u \mid \alpha)$ is finite. The $M$-th order moment of $Y \sim f(y \mid \boldsymbol{X}^\top\boldsymbol{\beta}, \sigma, \alpha)$ is also finite.*

*Proof.* From Proposition 5, it holds that

$$
\begin{aligned}
\mathbb{E}_{U\sim f_*}[U^m] &= \int u^m \phi\left(r_\alpha(u)\right) du \\
&= \int (v + H_\alpha(v))^m \phi(v)(1 + h_\alpha(v)) dv \\
&\leq \int |v + H_\alpha(v)|^m |1 + h_\alpha(v)| \phi(v) dv \\
&\leq \int (|v| + |H_\alpha(v)|)^m (1 + |h_\alpha(v)|) \phi(v) dv \\
&\leq \int (|v| + \rho_\alpha|v|)^m (1 + \rho_\alpha) \phi(v) dv \\
&= (1 + \rho_\alpha)^{m+1} \int |v|^m \phi(v) dv \\
&= (1 + \rho_\alpha)^{m+1} \mathbb{E}_{V\sim\phi}[|V|^m] < \infty.
\end{aligned}
\tag{86}
$$

Furthermore, since $y = \sigma u + \mu$, it obviously holds that

$$\mathbb{E}_{Y \sim f}[Y^m] = \mathbb{E}_{U \sim f_*}\left[(\sigma U + \boldsymbol{X}^\top \boldsymbol{\beta})^m\right]$$

$$= \sum_{k=0}^{m} \binom{m}{k} \sigma^k \left(\boldsymbol{X}^\top \boldsymbol{\beta}\right)^{m-k} \mathbb{E}_{U \sim f_*}\left[U^k\right] < \infty. \tag{87}$$

$\square$

**Proposition 10.** *Suppose that Assumption 1 is satisfied. Then, the third-order derivative of the log-likelihood function $l(\boldsymbol{\theta} \mid \boldsymbol{X}, y)$, more precisely, $\left|\frac{\partial^3}{\partial \theta_{i_1} \partial \theta_{i_2} \partial \theta_{i_3}} l(\boldsymbol{\theta} \mid \boldsymbol{X}, y)\right|$ for any $(i_1, i_2, i_3) \in \{1, \dots, P+2\}^3$ are bounded by a polynomial of $|y|$.*

*Proof.* The scores of the log-likelihood function $l(\boldsymbol{\theta} \mid \boldsymbol{X}, y)$ is

$$\frac{\partial}{\partial \boldsymbol{\beta}} l(\boldsymbol{\theta} \mid \boldsymbol{X}, y) = \frac{\partial}{\partial u} \frac{\partial u}{\partial \mu} \frac{\partial \mu}{\partial \boldsymbol{\beta}} l_*(\alpha \mid u)$$

$$= \frac{1}{\sigma} r_\alpha(u) r'_\alpha(u) \boldsymbol{X}, \tag{88}$$

$$\frac{\partial}{\partial \sigma} l(\boldsymbol{\theta} \mid \boldsymbol{X}, y) = \frac{\partial}{\partial u} \frac{\partial u}{\partial \sigma} l_*(\alpha \mid u) + \frac{\partial}{\partial \sigma} \log \frac{1}{\sigma}$$

$$= \frac{u}{\sigma} r_\alpha(u) r'_\alpha(u) - \frac{1}{\sigma}, \tag{89}$$

$$\frac{\partial}{\partial \alpha} l(\boldsymbol{\theta} \mid \boldsymbol{X}, y) = \frac{\partial}{\partial \alpha} l_*(\alpha \mid u)$$

$$= -r_\alpha(u) \frac{\partial}{\partial \alpha} r_\alpha(u). \tag{90}$$

The second-order derivatives of (88), (89), and (90) with respect to $\boldsymbol{\theta}$ consist of

$$u, \ r_\alpha(u), \ r'_\alpha(u), \ r''_\alpha(u), \ r'''_\alpha(u), \ \frac{\partial}{\partial \alpha} r_\alpha(u), \ \frac{\partial^2}{\partial \alpha^2} r_\alpha(u), \ \frac{\partial^3}{\partial \alpha^3} r_\alpha(u), \ \frac{\partial}{\partial \alpha} r'_\alpha(u), \ \frac{\partial}{\partial \alpha} r''_\alpha(u), \ \frac{\partial^2}{\partial \alpha^2} r'_\alpha(u), \tag{91}$$

as terms involving $u$. From Proposition 6 and Proposition 8, the second to eighth functions of (91) are bounded by some sextic function of $u$. Consider the remaining last three functions of (91). We have

$$\frac{\partial}{\partial \alpha} r'_\alpha(u) = -r''_\alpha(u) \frac{\partial}{\partial \alpha} H_\alpha(v), \tag{92}$$

$$\frac{\partial}{\partial \alpha} r''_\alpha(u) = -r'''_\alpha(u) \frac{\partial}{\partial \alpha} H_\alpha(v), \tag{93}$$

$$\frac{\partial^2}{\partial \alpha^2} r'_\alpha(u) = r'''_\alpha(u) \left(\frac{\partial}{\partial \alpha} H_\alpha(v)\right)^2 - r''_\alpha(u) \frac{\partial^2}{\partial \alpha^2} H_\alpha(v). \tag{94}$$

From Proposition 6 and Proposition 8, the absolutes of the above three functions are bounded by some quintic function of $|u|$. Since $u = (y - \mu)/\sigma$ and Assumption 1, the proof is complete. $\square$

**Proposition 11.** *Let $\xi(q_\alpha(v)) := \eta(v)$. Then, we have*

$$\mathbb{E}_{U \sim f_*(u \mid \alpha)}[\xi(U)] = \mathbb{E}_{V \sim \phi(v)}[\eta(V)(1 + h_\alpha(V))]. \tag{95}$$

*Proof.* On the basis of integration by substitution,

$$\mathbb{E}_{U \sim f_0(u \mid \alpha)}[\xi(U)] = \int \xi(u) \phi(r_\alpha(u)) du$$

$$= \int \eta(v) \phi(v)(1 + h_\alpha(v)) dv$$

$$= \mathbb{E}_{V \sim \phi(v)}[\eta(V)(1 + h_\alpha(V))]. \tag{96}$$

$\square$

For a condition $\mathcal{A}$, let $\mathbb{1}\{\mathcal{A}\}$ be the indicator function, which means $\mathbb{1}\{\mathcal{A}\} = 1$ if $\mathcal{A}$ is true and $\mathbb{1}\{\mathcal{A}\} = 0$ if $\mathcal{A}$ is false.

**Proposition 12.** *For $V \sim \mathcal{N}(0,1)$ and a sufficiently large positive constant $M$, it holds that*

$$
\mathbb{E}\left[|V|^m \, \mathbb{1}\{|V| > M\}\right] \leq
\begin{cases}
\exp\left(-\frac{M^2}{2}\right) & (m = 0) \\
\left(M^{m-1} + O\left(M^{m-2}\right)\right)\exp\left(-\frac{M^2}{2}\right) & (m = 1, 2, 3, 4)
\end{cases}
\tag{97}
$$

*Proof.* For $m = 0$,

$$
\begin{aligned}
\mathbb{E}\left[\mathbb{1}\{|V| > M\}\right] &= 2\int_M^\infty \phi(v)dv \\
&= 2\int_0^\infty \phi(v + M)dv \\
&\leq 2\exp\left(-\frac{M^2}{2}\right)\int_0^\infty \phi(v)dv \\
&= \exp\left(-\frac{M^2}{2}\right).
\end{aligned}
\tag{98}
$$

For $m = 1, 2, 3, 4$,

$$
\begin{aligned}
\mathbb{E}\left[|V| \, \mathbb{1}\{|V| > M\}\right] &= 2\int_M^\infty v\phi(v)dv \\
&= \sqrt{\frac{2}{\pi}}\exp\left(-\frac{M^2}{2}\right) \\
&\leq \exp\left(-\frac{M^2}{2}\right),
\end{aligned}
\tag{99}
$$

$$
\begin{aligned}
\mathbb{E}\left[|V|^2 \, \mathbb{1}\{|V| > M\}\right] &= 2\int_M^\infty v^2\phi(v)dv \\
&= \sqrt{\frac{2}{\pi}}M\exp\left(-\frac{M^2}{2}\right) + \mathbb{E}\left[\mathbb{1}\{|V| > M\}\right] \\
&\leq \sqrt{\frac{2}{\pi}}M\exp\left(-\frac{M^2}{2}\right) + \exp\left(-\frac{M^2}{2}\right) \\
&\leq (M + 1)\exp\left(-\frac{M^2}{2}\right),
\end{aligned}
\tag{100}
$$

$$
\begin{aligned}
\mathbb{E}\left[|V|^3 \, \mathbb{1}\{|V| > M\}\right] &= 2\int_M^\infty v^3\phi(v)dv \\
&= \sqrt{\frac{2}{\pi}}M^2\exp\left(-\frac{M^2}{2}\right) + 2\mathbb{E}\left[|V| \, \mathbb{1}\{|V| > M\}\right] \\
&\leq \sqrt{\frac{2}{\pi}}M^2\exp\left(-\frac{M^2}{2}\right) + 2\exp\left(-\frac{M^2}{2}\right) \\
&\leq (M^2 + 2)\exp\left(-\frac{M^2}{2}\right),
\end{aligned}
\tag{101}
$$

$$
\begin{aligned}
\mathbb{E}\left[|V|^4 \, \mathbb{1}\{|V| > M\}\right] &= 2 \int_M^\infty v^4 \phi(v) dv \\
&= \sqrt{\frac{2}{\pi}} M^3 \exp\left(-\frac{M^2}{2}\right) + 3\mathbb{E}\left[|V|^2 \, \mathbb{1}\{|V| > M\}\right] \\
&\leq \sqrt{\frac{2}{\pi}} M^3 \exp\left(-\frac{M^2}{2}\right) + 3(M+1) \exp\left(-\frac{M^2}{2}\right) \\
&\leq (M^3 + 3M + 3) \exp\left(-\frac{M^2}{2}\right).
\end{aligned}
\tag{102}
$$

$\square$

**Proposition 13.** *For $U \sim f_*(u \mid \alpha)$ and a sufficiently large positive constant $M$, it holds that*

$$
\mathbb{E}\left[|U|^m \, \mathbb{1}\{|U| > M\}\right] \leq
\begin{cases}
2 \exp\left(-\frac{M^2}{8}\right) & (m = 0) \\
4\left(M^{m-1} + O\left(M^{m-2}\right)\right) \exp\left(-\frac{M^2}{8}\right) & (m = 1, 2, 3, 4)
\end{cases}
\tag{103}
$$

*Proof.* From Proposition 5, we have $|q_\alpha(v)| \leq |v| + |H_\alpha(v)| < 2|v|$ and $|1 + h_\alpha(v)| < 1 + |h_\alpha(v)| < 2$. In addition to this, using Proposition 11 and Proposition 12, it holds that

$$
\begin{aligned}
\mathbb{E}\left[|U|^m \, \mathbb{1}\{|U| > M\}\right] &= \mathbb{E}\left[|q_\alpha(V)|^m \, (1 + h_\alpha(V)) \mathbb{1}\{|q_\alpha(V)| > M\}\right] \\
&\leq 2^{m+1} \mathbb{E}\left[|V|^m \, \mathbb{1}\left\{|V| > \frac{M}{2}\right\}\right] \\
&=
\begin{cases}
2 \exp\left(-\frac{M^2}{8}\right) & (m = 0) \\
2^{m+1}\left(\left(\frac{M}{2}\right)^{m-1} + O\left(\left(\frac{M}{2}\right)^{m-2}\right)\right) \exp\left(-\frac{M^2}{8}\right) & (m = 1, 2, 3, 4)
\end{cases} \\
&=
\begin{cases}
2 \exp\left(-\frac{M^2}{8}\right) & (m = 0) \\
4\left(M^{m-1} + O\left(M^{m-2}\right)\right) \exp\left(-\frac{M^2}{8}\right) & (m = 1, 2, 3, 4)
\end{cases}
\end{aligned}
\tag{104}
$$

$\square$

**Proposition 14.** *For $Y \sim f(y \mid \zeta)$ and a sufficiently large positive constant $M$, it holds that*

$$
\mathbb{E}\left[|Y|^m \, \mathbb{1}\{|Y| > M\}\right] \leq
\begin{cases}
2 \exp\left(-\frac{1}{8}\left(\frac{M-K}{K}\right)^2\right) & (m = 0) \\
4K\left(M^{m-1} + O\left(M^{m-2}\right)\right) \exp\left(-\frac{1}{8}\left(\frac{M-K}{K}\right)^2\right) & (m = 1, 2, 3, 4)
\end{cases}
\tag{105}
$$

*where $K$ is based on the parameter space* (9).

*Proof.* Note that $|\mu| \leq K$ and $\sigma \leq K$ from (9). Using Proposition 13, it holds that

$$
\begin{aligned}
\mathbb{E}\left[|Y|^m \, \mathbb{1}\{|Y| > M\}\right] &= \mathbb{E}\left[|\mu + \sigma U|^m \, \mathbb{1}\{|\mu + \sigma U| > M\}\right] \\
&\leq \mathbb{E}\left[(K + K|U|)^m \, \mathbb{1}\{K + K|U| > M\}\right] \\
&= K^m \sum_{i=0}^m {}_m\mathrm{C}_i \mathbb{E}\left[|U|^i \, \mathbb{1}\left\{|U| > \frac{M-K}{K}\right\}\right] \\
&\leq 2K^m \exp\left(-\frac{1}{8}\left(\frac{M-K}{K}\right)^2\right) \\
&\quad + 4K^m \sum_{i=1}^m {}_m\mathrm{C}_i \left(\left(\frac{M-K}{K}\right)^{i-1} + O\left(\left(\frac{M-K}{K}\right)^{i-2}\right)\right) \exp\left(-\frac{1}{8}\left(\frac{M-K}{K}\right)^2\right) \\
&=
\begin{cases}
2 \exp\left(-\frac{1}{8}\left(\frac{M-K}{K}\right)^2\right) & (m = 0) \\
4K\left(M^{m-1} + O\left(M^{m-2}\right)\right)\left(-\frac{1}{8}\left(\frac{M-K}{K}\right)^2\right) & (m = 1, 2, 3, 4)
\end{cases}
\end{aligned}
\tag{106}
$$

$\square$

**Proposition 15.** *Let the upper bound of* (12) *be denoted by* $G_5(y)$. *Let*

$$G(y \mid M) := G_5(y)\mathbb{1}\{G_5(y) > M\}. \tag{107}$$

*Then, for* $Y \sim f(y \mid \boldsymbol{\zeta})$ *and a sufficiently large positive constant* $M$, *there exist some positive constants* $C_{5,1}$, $C_{5,2}$, *and* $C_{5,3}$ *such that*

$$\mathbb{E}[G(Y \mid M)] \leq C_{5,1}M^{\frac{1}{2}}\exp(-C_{5,3}M), \tag{108}$$

$$\mathbb{E}[G^2(Y \mid M)] \leq C_{5,2}M^{\frac{3}{2}}\exp(-C_{5,3}M). \tag{109}$$

*Proof.* From Proposition 1, we know

$$G_5(y) = C_{1,2}|y|^2 + C_{1,1}|y| + C_{1,0} > M, \tag{110}$$

and then we have

$$|y| > \frac{-C_{1,1} + \sqrt{C_{1,1}^2 + 4C_{1,2}(M - C_{1,0})}}{2C_{1,2}} \geq \sqrt{C_{5,4}M} \tag{111}$$

where $C_{5,4}$ is some positive constant. From Proposition 14,

$$\begin{aligned}
\mathbb{E}[G(Y \mid M)] &\leq \mathbb{E}\left[\left(C_{1,2}|Y|^2 + C_{1,1}|Y| + C_{1,0}\right)\mathbb{1}\left\{|Y| > \sqrt{C_{5,4}M}\right\}\right] \\
&\leq 4C_{1,2}K\left(\sqrt{C_{5,4}M} + O(1)\right)\exp\left(-\frac{1}{8}\left(\frac{\sqrt{C_{5,4}M} - K}{K}\right)^2\right) \\
&\leq C_{5,1}M^{\frac{1}{2}}\exp(-C_{5,3}M),
\end{aligned} \tag{112}$$

where $C_{5,1}$ and $C_{5,2}$ are some positive constants.

$$\begin{aligned}
\mathbb{E}[G^2(Y \mid M)] &\leq \mathbb{E}\left[\left(C_{1,2}|y|^2 + C_{1,1}|y| + C_{1,0}\right)^2\mathbb{1}\left\{|Y| > \sqrt{C_{5,4}M}\right\}\right] \\
&\leq 4C_{1,2}^2K\left(\left(\sqrt{C_{5,4}M}\right)^3 + O(M)\right)\exp\left(-\frac{1}{8}\left(\frac{\sqrt{C_{5,4}M} - K}{K}\right)^2\right) \\
&\leq C_{5,2}M^{\frac{3}{2}}\exp(-C_{5,3}M),
\end{aligned} \tag{113}$$

where $C_{5,2}$ is some positive constant. $\square$

**Proposition 16.** *Let* $G(y)$ *be the function defined by* (107). *Let*

$$F(y \mid M) := G(y \mid M) + \mathbb{E}[G(Y \mid M) \mid \boldsymbol{X}]. \tag{114}$$

*Let* $M_N := \frac{3}{C_{5,3}}\log N$ *for* $N \geq \exp\left(\frac{27C_{5,2}^2}{C_{5,3}^3}\right) \vee \left(\frac{3C_{5,1}^2}{C_{5,3}}\right)^{\frac{1}{4}} \vee e$. *Then, it holds that*

$$\mathbb{P}\left(\frac{1}{N}\sum_{n=1}^{N}F(Y_n \mid M_N) > \frac{2\log N}{N}\right) \leq \frac{1}{N}. \tag{115}$$

*Proof.* From $N \geq \exp\left(\frac{27C_{5,2}^2}{C_{5,3}^3}\right) \vee \left(\frac{3C_{5,1}^2}{C_{5,3}}\right)^{\frac{1}{4}} \vee e$, we have

$$\frac{3^{\frac{3}{2}}C_{5,2}}{C_{5,3}^{\frac{3}{2}}} \leq \sqrt{\log N} \leq 2\sqrt{\log N} - \frac{\sqrt{3}C_{5,1}}{\sqrt{C_{5,3}}N^2}. \tag{116}$$

Therefore, using Proposition 15, it holds

$$
\begin{aligned}
\mathbb{P}\left(\frac{1}{N}\sum_{n=1}^{N}F(Y_n \mid M_N) > \frac{2\log N}{N}\right) &\leq \mathbb{P}\left(\frac{1}{N}\sum_{n=1}^{N}\left(G(Y_n \mid M_N) + \mathbb{E}\left[G(Y \mid M_N \mid \boldsymbol{X}_n)\right]\right) > \frac{2\log N}{N}\right)\\
&= \mathbb{P}\left(\frac{1}{N}\sum_{n=1}^{N}G(Y_n \mid M_N) > \frac{2\log N}{N} - \frac{1}{N}\sum_{n=1}^{N}\mathbb{E}\left[G(Y \mid M_N) \mid \boldsymbol{X}_n\right]\right)\\
&\leq \mathbb{P}\left(\frac{1}{N}\sum_{n=1}^{N}G(Y_n \mid M_N) > \frac{2\log N}{N} - C_{5,1}M_N^{\frac{1}{2}}\exp(-C_{5,3}M_N)\right)\\
&= \mathbb{P}\left(\frac{1}{N}\sum_{n=1}^{N}G(Y_n \mid M_N) > \frac{\sqrt{\log N}}{N}\left(2\sqrt{\log N} - \frac{\sqrt{3}C_{5,1}}{\sqrt{C_{5,3}}N^2}\right)\right)\\
&\leq \mathbb{P}\left(\frac{1}{N}\sum_{n=1}^{N}G(Y_n \mid M_N) > \frac{\log N}{N}\right)\\
&\leq \mathbb{E}\left[G^2(Y \mid M_N)\right]\left(\frac{N}{\log N}\right)^2\\
&\leq C_{5,2}M_N^{\frac{3}{2}}\exp(-C_{5,3}M_N)\left(\frac{N}{\log N}\right)^2\\
&= \frac{3^{\frac{3}{2}}C_{5,2}}{C_{5,3}^{\frac{3}{2}}}\frac{1}{N\sqrt{\log N}}\\
&\leq \frac{1}{N}
\end{aligned}
\tag{117}
$$

$\square$

## E.2 Proof of Proposition 1

*Proof.* Note that $u = (y - \boldsymbol{X}^\top\boldsymbol{\beta})/\sigma$. This proposition is derived immediately from Assumption 1, (88), (89), (90), Proposition 6, and Proposition 8. $\square$

## E.3 Proof of Proposition 2

*Proof.* For simplicity of notation, let $\omega(v) := \sigma\frac{\partial}{\partial\alpha}H_\alpha(v)$. From the scores (88), (89), and (90), the Fisher information matrix can be described as follows by using Proposition 11:

$$
\begin{aligned}
\mathbb{E}\left[-\frac{\partial^2}{\partial\boldsymbol{\psi}\partial\boldsymbol{\psi}^\top}l(\boldsymbol{\psi} \mid Y)\right] &= \frac{1}{\sigma^2}\mathbb{E}_{V\sim\phi(v)}\left[\frac{1}{1+h_\alpha(V)}\begin{bmatrix} V^2 & V^2 H_\alpha & -V^2\omega \\ V^2 H_\alpha & V^2 H_\alpha^2 + V^4 & -V^2 H_\alpha\omega \\ -V^2\omega & -V^2 H_\alpha\omega & V^2\omega^2 \end{bmatrix}\right] - \frac{1}{\sigma^2}\begin{bmatrix} 0 & 0 & 0 \\ 0 & 1 & 0 \\ 0 & 0 & 0 \end{bmatrix}\\
&\succeq \frac{1}{2\sigma^2}\mathbb{E}_{V\sim\phi(v)}\begin{bmatrix} V^2 & V^2 H_\alpha & -V^2\omega \\ V^2 H_\alpha & V^2 H_\alpha^2 + 3 & -V^2 H_\alpha\omega \\ -V^2\omega & -V^2 H_\alpha\omega & V^2\omega^2 \end{bmatrix} - \frac{1}{\sigma^2}\begin{bmatrix} 0 & 0 & 0 \\ 0 & 1 & 0 \\ 0 & 0 & 0 \end{bmatrix}\\
&= \frac{1}{2\sigma^2}\mathbb{E}_{V\sim\phi(v)}\begin{bmatrix} V^2 & V^2 H_\alpha & -V^2\omega \\ V^2 H_\alpha & V^2 H_\alpha^2 + 1 & -V^2 H_\alpha\omega \\ -V^2\omega & -V^2 H_\alpha\omega & V^2\omega^2 \end{bmatrix}\\
&= \frac{1}{2\sigma^2}\mathbb{E}_{V\sim v^2\phi(v)}\begin{bmatrix} 1 & H_\alpha & -\omega \\ H_\alpha & H_\alpha^2 + 1 & -H_\alpha\omega \\ -\omega & -H_\alpha\omega & \omega^2 \end{bmatrix}.
\end{aligned}
\tag{118}
$$

Hereafter, since $v^2\phi(v)$ is a probability density function on $v \in \mathbb{R}$, the random variable $V$ is regarded as $V \sim v^2\phi(v)$ for notational simplicity. Note that matrix (118) is clearly positive semidefinite, we only need

to check that its determinant is positive. For $\alpha \to 0$, we have

$$
\begin{aligned}
\lim_{\alpha \to 0} H_\alpha(v) &= \lim_{\alpha \to 0} \rho_\alpha \frac{\alpha v^2}{\sqrt{1 + \alpha^2 v^2} + 1} \\
&= 0,
\end{aligned}
\tag{119}
$$

$$
\begin{aligned}
\lim_{\alpha \to 0} \omega(v) &= \sigma \lim_{\alpha \to 0} \left( \exp(-\alpha^2) \left( \sqrt{1 + \alpha^2 v^2} - 1 \right) + \rho_\alpha \frac{v^2}{\sqrt{1 + \alpha^2 v^2} \left( \sqrt{1 + \alpha^2 v^2} + 1 \right)} \right) \\
&= \frac{\sigma v^2}{4},
\end{aligned}
\tag{120}
$$

and then we see that the determinant of (118) is positive because

$$
\begin{aligned}
\det \left( \mathbb{E}_{V \sim v^2 \phi(v)} \begin{bmatrix} 1 & H_\alpha & -\omega \\ H_\alpha & H_\alpha^2 + 1 & -H_\alpha \omega \\ -\omega & -H_\alpha \omega & \omega^2 \end{bmatrix} \right) &= \begin{vmatrix} 1 & 0 & -\frac{\sigma}{4} \mathbb{E}_{V \sim v^2 \phi(v)} \left[ V^2 \right] \\ 0 & 1 & 0 \\ -\frac{\sigma}{4} \mathbb{E}_{V \sim v^2 \phi(v)} \left[ V^2 \right] & 0 & \frac{\sigma^2}{16} \mathbb{E}_{V \sim v^2 \phi(v)} \left[ V^4 \right] \end{vmatrix} \\
&= \frac{\sigma^2}{16} \left( \mathbb{E}_{V \sim v^2 \phi(v)} \left[ V^4 \right] - \mathbb{E}_{V \sim v^2 \phi(v)}^2 \left[ V^2 \right] \right) \\
&= \frac{3}{4} \sigma^2 \\
&> 0.
\end{aligned}
\tag{121}
$$

Consider the case $\alpha \neq 0$. We have

$$
\begin{aligned}
\det \left( \mathbb{E}_{V \sim v^2 \phi(v)} \begin{bmatrix} 1 & H_\alpha & -\omega \\ H_\alpha & H_\alpha^2 + 1 & -H_\alpha \omega \\ -\omega & -H_\alpha \omega & \omega^2 \end{bmatrix} \right) &= \begin{vmatrix} 1 & \mathbb{E}[H_\alpha] & -\mathbb{E}[\omega] \\ \mathbb{E}[H_\alpha] & \mathbb{E}[H_\alpha^2] + 1 & -\mathbb{E}[H_\alpha \omega] \\ -\mathbb{E}[\omega] & -\mathbb{E}[H_\alpha \omega] & \mathbb{E}[\omega^2] \end{vmatrix} \\
&= \left( \mathbb{E}\left[ H_\alpha^2 \right] \mathbb{E}\left[ \omega^2 \right] - \mathbb{E}\left[ H_\alpha^2 \right] \mathbb{E}^2 \left[ \omega \right] - \mathbb{E}^2 \left[ H_\alpha \right] \mathbb{E}\left[ \omega^2 \right] \right) \\
&\quad + 2\mathbb{E}[H_\alpha \omega] \mathbb{E}[H_\alpha] \mathbb{E}[\omega] - \mathbb{E}^2 [H_\alpha \omega] + \left( \mathbb{E}\left[ \omega^2 \right] - \mathbb{E}^2 \left[ \omega \right] \right) \\
&> -\mathbb{E}^2 [H_\alpha] \mathbb{E}^2 [\omega] + 2\mathbb{E}[H_\alpha \omega] \mathbb{E}[H_\alpha] \mathbb{E}[\omega] - \mathbb{E}^2 [H_\alpha \omega] + \mathbb{V}[\omega] \\
&= - \left( \mathbb{E}[H_\alpha \omega] - \mathbb{E}[H_\alpha] \mathbb{E}[\omega] \right)^2 + \mathbb{V}[\omega] \\
&= \mathbb{V}[\omega] - \mathrm{Cov}^2 [H_\alpha, \omega],
\end{aligned}
\tag{122}
$$

where the inequality is based on $\mathbb{V}[H_\alpha] \mathbb{V}[\omega] = \left( \mathbb{E}\left[ H_\alpha^2 \right] - \mathbb{E}^2 [H_\alpha] \right) \left( \mathbb{E}\left[ \omega^2 \right] - \mathbb{E}^2 [\omega] \right) > 0$. To examine the sign of (122), we evaluate $\mathbb{V}[H_\alpha]$ concretely, starting with (4), i.e.,

$$
\begin{aligned}
\mathbb{V}_{V \sim v^2 \phi(v)} [H_\alpha] &= \rho_\alpha^2 \mathbb{V}_{V \sim v^2 \phi(v)} \left[ \sqrt{V^2 + \frac{1}{\alpha^2}} \right] \\
&= \rho_\alpha^2 \left( \mathbb{E}_{V \sim v^2 \phi(v)} \left[ V^2 + \frac{1}{\alpha^2} \right] - \mathbb{E}_{V \sim v^2 \phi(v)}^2 \left[ \sqrt{V^2 + \frac{1}{\alpha^2}} \right] \right) \\
&= \rho_\alpha^2 \left( 3 + \frac{1}{\alpha^2} - \mathbb{E}_{V \sim v^2 \phi(v)}^2 \left[ \sqrt{V^2 + \frac{1}{\alpha^2}} \right] \right).
\end{aligned}
\tag{123}
$$

Here, we focus on the behavior of $\mathbb{E}_{V \sim v^2 \phi(v)} \left[ \sqrt{V^2 + \frac{1}{\alpha^2}} \right]$ with respect to $\alpha$. We have

$$
\frac{\partial}{\partial \alpha} \frac{1}{\sqrt{2 + \frac{1}{\alpha^2}}} \mathbb{E}_{V \sim v^2 \phi(v)} \left[ \sqrt{V^2 + \frac{1}{\alpha^2}} \right] = \frac{\partial}{\partial \alpha} \int_{\mathbb{R}} \sqrt{\frac{1 + \alpha^2 v^2}{1 + 2\alpha^2}} v^2 \phi(v) dv
$$

$$
= \int_{\mathbb{R}} \frac{\partial}{\partial \alpha} \sqrt{\frac{1 + \alpha^2 v^2}{1 + 2\alpha^2}} v^2 \phi(v) dv
$$

$$
= \frac{\alpha}{(1 + 2\alpha^2)^{\frac{3}{2}}} \int_{\mathbb{R}} \left( \frac{v^4}{\sqrt{1 + \alpha^2 v^2}} - \frac{2v^2}{\sqrt{1 + \alpha^2 v^2}} \right) \phi(v) dv
$$

$$
= \frac{\alpha}{(1 + 2\alpha^2)^{\frac{3}{2}}} \int_{\mathbb{R}} \left( \frac{v^2}{\sqrt{1 + \alpha^2 v^2}} - \frac{\alpha^2 v^4}{(1 + \alpha^2 v^2)^{\frac{3}{2}}} \right) \phi(v) dv
$$

$$
= \frac{\alpha}{(1 + 2\alpha^2)^{\frac{3}{2}}} \int_{\mathbb{R}} \frac{v^2}{(1 + \alpha^2 v^2)^{\frac{3}{2}}} \phi(v) dv, \tag{124}
$$

where the fourth equality is derived by

$$
\int_{\mathbb{R}} \frac{v^4}{\sqrt{1 + \alpha^2 v^2}} \phi(v) dv = - \int_{\mathbb{R}} \frac{v^3}{\sqrt{1 + \alpha^2 v^2}} (\phi(v))' dv
$$

$$
= -2 \underbrace{\lim_{R \to \infty} \frac{R^3 \phi(R)}{\sqrt{1 + \alpha^2 R^2}}}_{=0} + \int_{\mathbb{R}} \left( \frac{3v^2}{\sqrt{1 + \alpha^2 v^2}} - \frac{\alpha^2 v^4}{(1 + \alpha^2 v^2)^{\frac{3}{2}}} \right) \phi(v) dv. \tag{125}
$$

Hence, $\frac{\partial}{\partial \alpha} \frac{1}{2 + \frac{1}{\alpha^2}} \mathbb{E}^2_{V \sim v^2 \phi(v)} \left[ \sqrt{V^2 + \frac{1}{\alpha^2}} \right]$ is monotonically increasing function of $\alpha^2$. Moreover,

$$
\frac{1}{2 + \frac{1}{\alpha^2}} \mathbb{E}^2_{V \sim v^2 \phi(v)} \left[ \sqrt{V^2 + \frac{1}{\alpha^2}} \right] > \lim_{\alpha^2 \to 0} \frac{1}{2 + \frac{1}{\alpha^2}} \mathbb{E}^2_{V \sim v^2 \phi(v)} \left[ \sqrt{V^2 + \frac{1}{\alpha^2}} \right]
$$

$$
= 1. \tag{126}
$$

Thus, we have

$$
\mathbb{E}^2_{V \sim v^2 \phi(v)} \left[ \sqrt{V^2 + \frac{1}{\alpha^2}} \right] > 2 + \frac{1}{\alpha^2}, \tag{127}
$$

and $\mathbb{V}_{V \sim v^2 \phi(v)} [H_\alpha] < \rho_\alpha^2 < 1$ from (123). For $\alpha \neq 0$, $H_\alpha(v)$ and $\omega(v)$ aren't obviously constant functions. Then, $\mathbb{V}_{V \sim v^2 \phi(v)} [\omega(V)] < \infty$ because the upper bound of $\left| \frac{\partial}{\partial \alpha} H_\alpha(v) \right|$ is a linear function of $|u|$ from Proposition 7. Therefore, since $0 < \mathbb{V}[H_\alpha] < 1$ and $0 < \mathbb{V}[\omega] < \infty$, it holds that

$$
\mathrm{Cov}^2 [H_\alpha, \omega] < \mathbb{V}[H_\alpha] \mathbb{V}[\omega] < \mathbb{V}[\omega]. \tag{128}
$$

From (122) and (128), we see that $\mathcal{I}(\psi) = \mathbb{E} \left[ \frac{\partial^2}{\partial \psi \partial \psi^\top} l(\psi \mid Y) \right]$ is positive definite for any $\psi$. Consider the case $\psi = \psi^0(X) = (X^\top \beta^0, \sigma^0, \alpha^0)$. Since $\psi^0(X)$ belongs to the bounded set $\Psi$, we have $\inf_X \Lambda_{\min}(\mathcal{I}(\psi^0(X))) > 0$. $\qquad \square$

### E.4 Proof of Proposition 3

*Proof.* From Proposition 10, with some natural number $M$ and some positive constant $C_{3,m}$ for $m = 0, 1, \ldots, M$, we have

$$
\max_{(i_1, i_2, i_3) \in \{1,2,3\}^3} \left| \frac{\partial^3}{\partial \psi_{i_1} \partial \psi_{i_2} \partial \psi_{i_3}} l(\boldsymbol{\theta} \mid \boldsymbol{X}, y) \right| \leq \sum_{m=0}^{M} C_{3,m} |y|^m. \tag{129}
$$

Since the $m$-th order moment of $Y \sim f(y \mid \boldsymbol{X}^\top \boldsymbol{\beta}, \sigma, \alpha)$ is finite from Proposition 9, the proof is complete. $\qquad \square$

### E.5   Proof of Proposition 4

*Proof.* Applying the Taylor expansion around the point $\boldsymbol{\psi}^0$ for (15), there exists some parameter vector $\tilde{\boldsymbol{\psi}} \in \{t\boldsymbol{\psi}^0 + (1-t)\boldsymbol{\psi} \in \mathbb{R}^3 \mid t \in [0,1]\}$ such that

$$\mathcal{E}(\boldsymbol{\psi} \mid \boldsymbol{\psi}^0) = \mathcal{E}(\boldsymbol{\psi}^0 \mid \boldsymbol{\psi}^0) + \left.\frac{\partial \mathcal{E}}{\partial \boldsymbol{\psi}^\top}\right|_{\boldsymbol{\psi}=\boldsymbol{\psi}^0} (\boldsymbol{\psi} - \boldsymbol{\psi}^0) + \frac{1}{2}(\boldsymbol{\psi} - \boldsymbol{\psi}^0)^\top \left.\frac{\partial^2 \mathcal{E}}{\partial \boldsymbol{\psi} \partial \boldsymbol{\psi}^\top}\right|_{\boldsymbol{\psi}=\tilde{\boldsymbol{\psi}}} (\boldsymbol{\psi} - \boldsymbol{\psi}^0). \tag{130}$$

Excess Risk (15) satisfies $\mathcal{E}(\boldsymbol{\psi}^0 \mid \boldsymbol{\psi}^0) = 0$ and $\frac{\partial \mathcal{E}}{\partial \boldsymbol{\psi}}(\boldsymbol{\psi}^0 \mid \boldsymbol{\psi}^0) = 0$ by its definition. Because $\mathcal{I}(\tilde{\boldsymbol{\psi}})$ is positive definite from Proposition 2, its smallest eigenvalue $\Lambda_{\min}(\mathcal{I}(\tilde{\boldsymbol{\psi}}))$ is positive. Thus, the quadratic term of (130) is evaluated as

$$(\boldsymbol{\psi} - \boldsymbol{\psi}^0)^\top \left.\frac{\partial^2 \mathcal{E}}{\partial \boldsymbol{\psi} \partial \boldsymbol{\psi}^\top}\right|_{\boldsymbol{\psi}=\tilde{\boldsymbol{\psi}}} (\boldsymbol{\psi} - \boldsymbol{\psi}^0) \geq \Lambda_{\min}(\mathcal{I}(\tilde{\boldsymbol{\psi}}))\|\boldsymbol{\psi} - \boldsymbol{\psi}^0\|_2^2. \tag{131}$$

Therefore, letting $C_2 := \inf_{X \in \mathcal{X}} \Lambda_{\min}(\mathcal{I}(\tilde{\boldsymbol{\psi}}(\boldsymbol{X}))) > 0$, based on

$$\begin{aligned}
\inf_{X \in \mathcal{X}} \inf_{\boldsymbol{\psi} \in \Theta: \|\boldsymbol{\psi} - \boldsymbol{\psi}^0\|_2 > \epsilon} \mathcal{E}(\boldsymbol{\psi} \mid \boldsymbol{\psi}^0) &\geq \inf_{X \in \mathcal{X}} \inf_{\boldsymbol{\psi} \in \Theta: \|\boldsymbol{\psi} - \boldsymbol{\psi}^0\|_2 > \epsilon} \frac{1}{2}\Lambda_{\min}(\mathcal{I}(\tilde{\boldsymbol{\psi}}(\boldsymbol{X})))\|\boldsymbol{\psi} - \boldsymbol{\psi}^0\|_2^2 \\
&> \frac{\epsilon^2}{2} \inf_{X \in \mathcal{X}} \Lambda_{\min}(\mathcal{I}(\tilde{\boldsymbol{\psi}}(\boldsymbol{X}))) \\
&> \frac{\epsilon^2 C_2}{2}, \tag{132}
\end{aligned}$$

we can take $\delta_\epsilon = \frac{\epsilon^2 C_2}{2} > 0$ as $\delta_\epsilon$ satisfying Proposition 4. $\square$

## F   Proof of Theorem 3

If Condition 1, 2, and 3 in Section 9.4.1.1 of Bühlmann & Van De Geer (2011) holds, the proposition holds immediately from Lemma 9.1 of Bühlmann & Van De Geer (2011). These Condition 1, 2, and 3 correspond to Proposition 3, 2, and 4, which were already shown, respectively. The proof is complete.

## G   Proof of Theorem 4

If Condition 1, 2, and 3 in Section 9.4.1.1 and Condition 4 in Section 9.4.2 of Bühlmann & Van De Geer (2011) holds, the proposition holds immediately from Theorem 9.1 of Bühlmann & Van De Geer (2011). These Condition 1, 2, and 3 correspond to Proposition 3, 2, and 4, which were already shown, respectively. Condition 4 correspond to Assumption 2. The proof is complete.

## H   Proof of Theorem 5

Theorem 5 is similar to Corollary 9.1 of Bühlmann & Van De Geer (2011). The difference is a model. The former model is a regression model with the mode-invariant skew-normal noise, which belongs to a class of generalized linear models, but the latter model is a finite mixture model. The proof of Corollary 9.1 of Bühlmann & Van De Geer (2011) is based on two parts; one is a model-specific part, and the other is not related to a model. The model-specific part consists of Proposition 9.4 and Lemma 9.3 of Bühlmann & Van De Geer (2011). We prove these propositions for generalized linear models. Using them, we can apply the same proof technique as in Bühlmann & Van De Geer (2011) to prove Theorem 5.

Proposition 9.4 and Lemma 9.3 of Bühlmann & Van De Geer (2011) show how to obtain an upper bound of the score function and how to set $M_N$. For our sparse regression model with the mode-invariant skew-normal noise, the upper bound of the score function is obtained in Proposition 1, which corresponds to Proposition 9.4 of Bühlmann & Van De Geer (2011). In this case, following the approach described in Bühlmann & Van De Geer (2011), we set $M_N = O(\log N)$, and then we can derive Proposition 16, which corresponds to Lemma 9.3 of Bühlmann & Van De Geer (2011). The proof is complete.

# I   Additional Experimental Details and Results

## I.1   Definition of Sparsity

In Section 4, we used the measure for sparsity (denoted by "Sparsity"), as introduced by Hurley & Rickard (2009). In this section, we briefly review its definition.

Given a vector $\boldsymbol{\beta} \in \mathbb{R}^P$, we sort the coefficients from the smallest to largest, i.e., $\beta_{(1)} \leq \ldots \leq \beta_{(P)}$, where $(p)$ is the $p$-th new index after sorting. Then, the "Sparsity" of $\boldsymbol{\beta}$ is defined by

$$\mathrm{Sparsity}(\boldsymbol{\beta}) = 1 - \frac{2}{\|\boldsymbol{\beta}\|_1} \sum_{p=1}^{P} |\beta_{(p)}| \left( \frac{P - p + \frac{1}{2}}{P} \right). \tag{133}$$

This measure is based on a weighted sum of all coefficients, evaluating how important a particular coefficient is to an overall sparsity. We have $\mathrm{Sparsity}(\boldsymbol{\beta}) = 0$ for the least sparse case where all coefficients have the same value, and $\mathrm{Sparsity}(\boldsymbol{\beta}) = 1$ for the most sparse case where there exist few non-zero coefficients under the situation $P \to \infty$. Therefore, the measure can evaluate sparsity of $\boldsymbol{\beta}$.

## I.2   Comparison with Other Methods

Adding to Section 4.1, we also compared the proposed model with a linear regression model without regularization, assuming the Azzalini's skew-normal distribution, skew-$t$ distribution, and skew-Cauchy distribution for noise (Arellano-Valle & Azzalini, 2013; Azzalini & Arellano-Valle, 2013; Azzalini & Salehi, 2020). These methods are denoted by "skew-N," "skew-t," and "skew-C," respectively. The experimental setting and data are the same as in Section 4.1.

Four evaluation measures, i.e., $\mathrm{MSE}(\hat{y})$, $\mathrm{MSE}(\hat{\boldsymbol{\beta}})$, $\mathrm{Sparsity}(\hat{\boldsymbol{\beta}})$, and Model $\mathrm{Size}(\hat{\boldsymbol{\beta}})$, are shown in Figure 5. The definitions of these measures are the same as in Section 4.1. As a result of Figure 5, the proposed method outperformed the comparative methods in all cases, indicating better prediction with a smaller number of features. Note that the model size for "skew-N," "skew-t," and "skew-C" is always 100 due to no regularization term, and the results of "Proposed," "Chen+," and "Lasso" are the same as in Figure 2.

## I.3   Real-World Financial Data

We applied the proposed method to the Engineering Graduate Salary (**EGS**) prediction data (Aggarwal et al., 2016), which provides engineering graduates' employment outcomes with standardized assessment scores. EGS has fewer features than the two medical datasets in Section 4.3, and the meanings of all features are specifically given. These properties allow us to consider the validity of the estimated active features. For simplicity, we used only numerical type features (see Table 4 for details). The data used here consisted of $N = 3998$ and $P = 25$. Other experimental conditions were the same as in Section 4.3.

Table 3 shows the result of EGS with the means and standard deviations (in parentheses) of the evaluation measures. The proposed method simultaneously achieved the smallest prediction error and the smallest model size. Regarding the residuals of the test data after training with Lasso, the mean of the normalized skewness of the residuals was 2.24.

We also obtained interesting results. Figure 6 shows box plots of the estimated coefficients for each method. The top 10 coefficients are picked out based on the averaged absolute values over 30 trials. For example, a graduate's quantitative ability ("Quant") ranked first for the proposed method, while a graduation year ("12graduation") ranked first for Lasso and Lasso+YJ. In the proposed method, "12graduation" ranked fourth with about half an effect. It may sound strange that the most relevant factor for salary is the graduation year. As the box plots indicate, the coefficient of this feature was sometimes zero. Moreover, an English score ("English") and a college tier ("CollegeTier") ranked second and third in the proposed method. From these points, the proposed method could select more reasonable features.

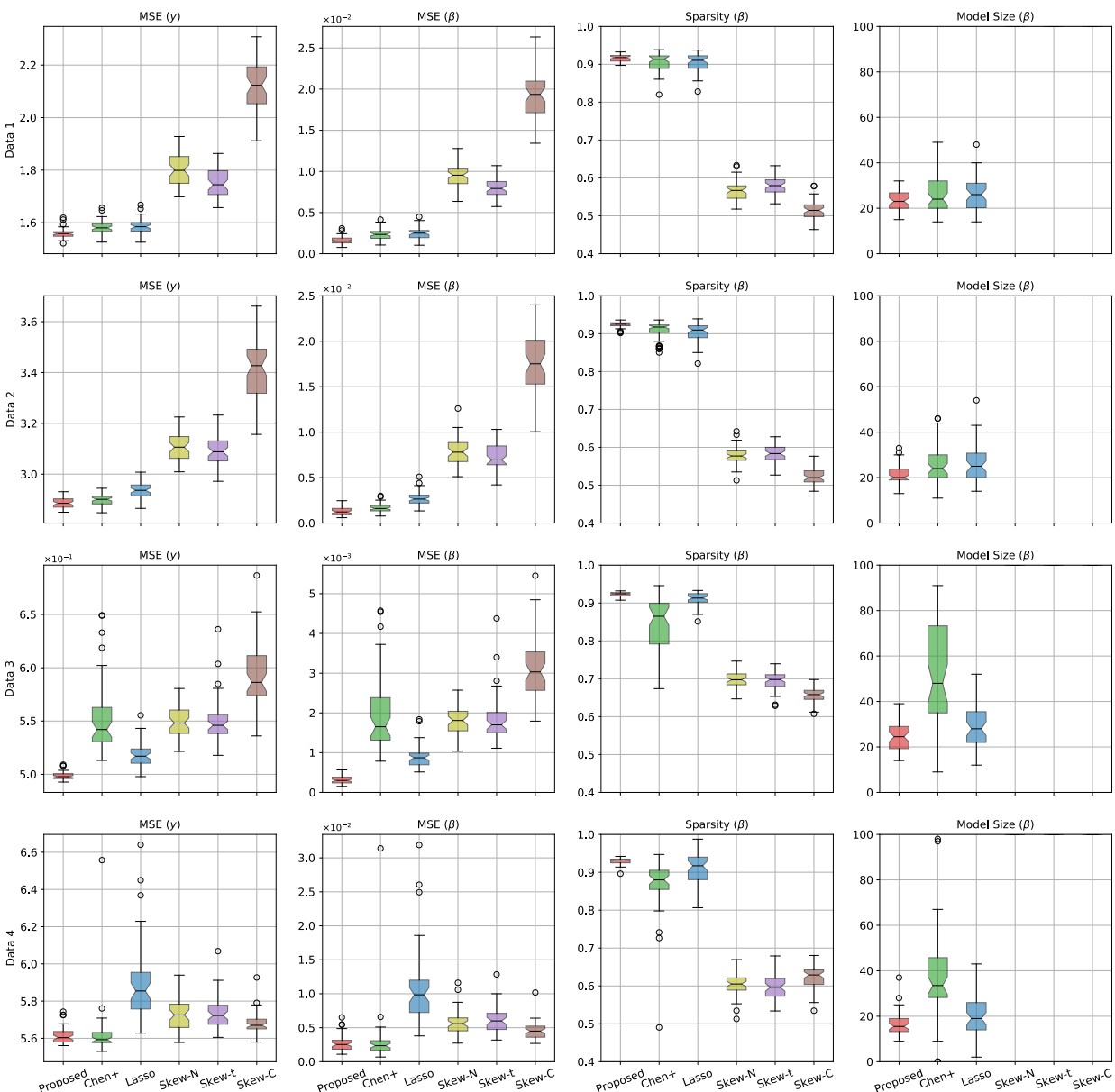

Figure 5: Comparison of six methods with four different types of simulation noises in terms of four evaluation measures. From top to bottom, each row results from Data 1, Data 2, Data 3, and Data 4, respectively. All experiments were conducted for 50 runs with different random seeds.

Table 3: Results of EGS ($N = 3998$, $P = 25$).

| | $\mathbf{MSE}(\hat{y})$ | $\mathbf{Sparsity}(\hat{\beta})$ | $\mathbf{Size}(\hat{\beta})$ |
|---|---|---|---|
| **Proposed** | $\mathbf{2.11 \times 10^1}$ $(3.14 \times 10^1)$ | $\mathbf{4.62 \times 10^{-1}}$ $(2.84 \times 10^{-2})$ | $\mathbf{2.31 \times 10^1}$ $(1.05 \times 10^0)$ |
| Chen+ | $4.97 \times 10^1$ $(7.59 \times 10^1)$ | $1.16 \times 10^{-1}$ $(16.8 \times 10^{-2})$ | $8.33 \times 10^1$ $(12.0 \times 10^0)$ |
| Lasso | $4.89 \times 10^1$ $(7.57 \times 10^1)$ | $3.78 \times 10^{-1}$ $(3.72 \times 10^{-2})$ | $2.44 \times 10^1$ $(0.68 \times 10^0)$ |
| Lasso + YJ | $2.64 \times 10^1$ $(3.86 \times 10^1)$ | $3.78 \times 10^{-1}$ $(3.72 \times 10^{-2})$ | $2.44 \times 10^1$ $(0.68 \times 10^0)$ |

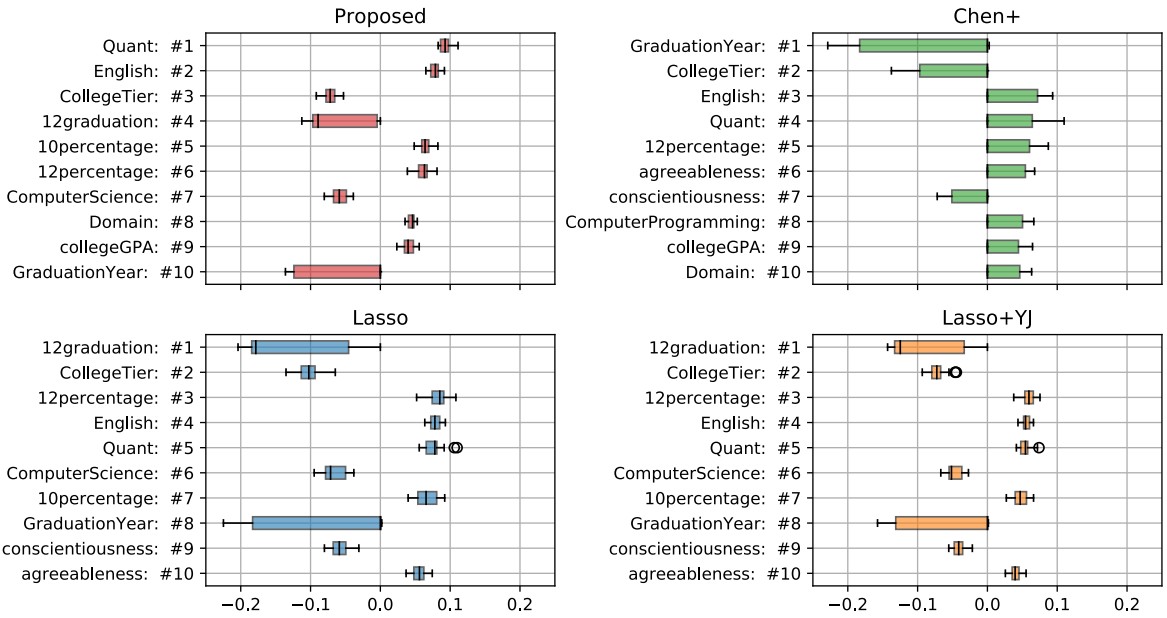

Figure 6: Box plots of the estimated coefficients for EGS. In the vertical axis, the top 10 coefficients are picked out based on the averaged absolute values over 30 runs with different random seeds. Details of each feature are described in Table 4.

Table 4: Description of features in EGS data (see Table 2 in Aggarwal et al. (2016) for details).

| Feature | Description |
|---:|:---|
| 10percentage | Overall marks obtained in grade 10 examinations |
| 12graduation | Year of graduation (senior year high school) |
| 12percentage | Overall marks obtained in grade 12 examinations |
| CollegeID | Unique ID identifying the college which the candidate attended |
| CollegeTier | Tier of college (computed from the average AMCAT scores) |
| CollegeGPA | Aggregate GPA at graduation |
| CollegeCityID | A unique ID to identify the city in which the college is located |
| CollegeCityTier | The tier of the city in which the college is located |
| GraduationYear | Year of graduation (Bachelor's degree) |
| English | Score in AMCAT's English section |
| Logical | Score in AMCAT's Logical ability section |
| Quant | Score in AMCAT's Quantitative ability section |
| Domain | Score in AMCAT's Domain module |
| ComputerProgramming | Score in AMCAT's Computer programming section |
| ElectronicsAndSemicon | Score in AMCAT's Electronics & Semiconductor Engineering section |
| ComputerScience | Score in AMCAT's Computer Science section |
| MechanicalEngg | Score in AMCAT's Mechanical Engineering section |
| ElectricalEngg | Score in AMCAT's Electrical Engineering section |
| TelecomEngg | Score in AMCAT's Telecommunication Engineering section |
| CivilEngg | Score in AMCAT's Civil Engineering section |
| conscientiousness | Score in one of the sections of AMCAT's personality test |
| agreeableness | Score in one of the sections of AMCAT's personality test |
| extraversion | Score in one of the sections of AMCAT's personality test |
| nueroticism | Score in one of the sections of AMCAT's personality test |
| openess_to_experience | Score in one of the sections of AMCAT's personality test |

