# OpenReview forum: "Sparse Modal Regression with Mode-Invariant Skew Noise"
_TMLR — Accepted by TMLR_

### Review · Reviewer_rUk6 · 2024-05-27

**Summary Of Contributions:**

This paper studies the $\ell_1$-penalized maximum-likelihood estimator for sparse linear regression with skewed noise. Whereas the same topic has been studied by Chen et al. (2014), this paper adopts a different noise model called the mode-invariant skew-normal distribution, proposed by Fujisawa and Abe (2015). A heuristic optimization method to compute the $\ell_1$-penalized maximum-likelihood (ML) estimate is provided. Both the estimation error and the variable screening ability of the estimator are analyzed.

**Audience:**

Yes

**Broader Impact Concerns:**

N/A.

**Claims And Evidence:**

Yes

**Requested Changes:**

Please address the weaknesses above.

**Strengths And Weaknesses:**

**Strengths.**
1. The result is novel and complete. The noise model considered is different from that in the previous work by Chen et al. (2014). Both the computational and statistical aspects are discussed.

**Weaknesses.**
1. The paper argues that the considered noise model is more interpretable. However, interpretability is a vague notion and lacks an objective definition. It would be beneficial if the authors could provide more objective, convincing motivation for their work.
2. The optimization method for computing the estimate follows a majorization-minimization argument. However, the method appears to be heuristic and lacks a convergence guarantee.
3. Theorem 4, which ensures that the guarantees of the estimation error and variable screening ability hold with high probability, is obtained by applying the results of Bühlmann & Van De Geer (2011), which were derived for the FMR model. The FMR model is a mixture-of-Gaussian linear regression model. It is unclear to me why the linear regression model with skewed noise, which this paper studies, is considered a special case of the FMR model.
4. Below are some minor comments.
    - The function $H_\alpha$ looks confusing in (3), as its definition appears far from (3) in the next paragraph.
    - Last line on p. 4: The word "bound" should be replaced with "bounded."
    - The purpose of Section 3.1 is unclear from the main text.
    - Second paragraph of Section 3.1: The meaning of the notation $\mathcal{X} = \{ X \}$ is unclear to me.
    - Assumption 2: Define $\kappa = 1 / \tilde{\kappa}$ can be confusing.

---

> ### Author Response · Authors · 2024-06-24
> **Response to Reviewer rUk6**
>
> We would like to thank you for providing many helpful comments. Your feedback on optimization methods has provided us with fresh perspectives and significantly enhanced the quality of our paper. We deeply appreciate your feedback.
>
>
>
> ### **Weakness 1**
> Our proposed method is an $\ell_1$ regularization regression method capable of regressing for a mode regardless of noise's skewness. This makes it easy to interpret the statistical role of sparse features, even under strong suspicion of noise's skewness. For these reasons, we say that our proposed method has high interpretability.
>
> However, as you pointed out, "interpretability" is widely used in various meanings. We have tried clarifying the explanation of "interpretability" in the revised paper.
>
> **Major Revisions:**
> - In the fourth paragraph of Section 1, we have added the sentence: "the proposed method can regress for its mode regardless of skewness, making it easy to interpret the statistical role of sparse features."
> - We have used "statistical interpretability" instead of the simple word "interpretability" and have explained its meaning.
>
>
>
> ### **Weakness 2**
> In response to your valuable comments, we have conducted a thorough study on the convergence of Algorithm 1. We have succeeded in obtaining a new theorem about the convergence of the update algorithm in which every cluster point in a sequence of $\theta$ is a stationary point.
>
> Before presenting a new theorem, we need to slightly change Algorithm 1. This algorithm updates $\sigma$ and $\alpha$ simultaneously, but we change it to update them alternately; more precisely, it updates $\sigma$ for a fixed $\beta$ and $\alpha$ and then updates $\alpha$ for a fixed $\beta$ and $\sigma$. (All numerical experiments were calculated using this implementation.)
>
> The new theorem about the convergence of the update algorithm is given as follows.
>
> **Theorem.** *Let $(\theta^{(t)})_{t=0,1,\ldots}$ be a sequence of $\theta$ after updating $\sigma$ in Algorithm 1.*
> *Then, every cluster point of $(\theta^{(t)})_{t=0,1,\ldots}$ is a stationary point.*
>
> This theorem can be proved using Mairal (2013) and Tseng (2001). Please see the revised paper for details.
>
> **Major Revisions:**
> - A new theorem (Theorem 2 in the revised paper) about the convergence of Algorithm 1 has been added.
> - Proof of the new theorem has been given in Appendix D.
> - In Section 2.3, the optimization method has been described in more detail than before.
>
> **References:**
> - Julien Mairal. Optimization with first-order surrogate functions. In *International Conference on Machine Learning*, pp. 783–791. PMLR, 2013.
> - Paul Tseng. Convergence of a block coordinate descent method for nondifferentiable minimization. *Journal of optimization theory and applications*, 109:475–494, 2001.
>
> ### **Weakness 3**
> Our proposed model differs from the FMR model and does NOT correspond to any special case of the FMR model. However, both models belong to the same superclass of generalized linear models. Two estimation procedures are based on non-convex negative log-likelihood, and hence, the proof techniques are essentially similar. The differences that impact the proof of Theorem 4 (Theorem 5 in the revised paper) are the upper bound of the score function and the setting of $M_N$, as detailed in Appendix H of the revised paper.
>
> - From Proposition 1, the score function of our proposed method is bounded by a second-order polynomial of $|y|$. In contrast, Proposition 9.4 of Buhlmann & Van De Geer (2011) bounds the score function of the FMR model by a first-order polynomial.
> - From Proposition 16, we can derive (104) with $M_N=O(\log N)$ for our proposed method. Lemma 9.3 of Buhlmann & Van De Geer (2011) allows us to obtain the corresponding inequality with $M_N=O(\sqrt{\log N})$ for the FMR model.
>
> These two differences are interconnected by Proposition 15.
>
>
>
> ### **Weakness 4**
> **Reply and Revisions:**
> - The function $H_{\alpha}$: Following your comment, we have placed the definition of $H_\alpha$ just before its specific formula (4).
> - Last line on p.4: We have revised it, following your comment.
> - The purpose of Section 3.1: In the first paragraph of Section 3, we have explained that the basic properties in Section 3.1 are used to prove subsequent theorems.
> - Second paragraph of Section 3.1: Although not indicated symbolically, $\mathcal{X}$ means some bounded space in $\mathbb{R}^P$.
> - Assumption 2: Following your comment, we have moved the definition $\kappa^{-1}:=\tilde{\kappa}(L,\mathcal{S}_0)$ into Theorem 4, which is actually used.

---

> > ### Comment · Reviewer_rUk6 · 2024-08-22
> >
> > I thank the authors for their clarifications and efforts to improve the paper. I apologize for my delayed response; I have recently been ill.
> >
> > 1. The proof of Theorem 2 utilizes a proposition from Mairal (2013). This lemma applies to so-called “first-order surrogates,” which must satisfy both majorization and smoothness properties (see Definition 2.1 of Mairal, 2013). However, the proof of Theorem 2 fails to verify both properties. Note that Line 8 in the proposed optimization method does not directly correspond to minimizing a “proximal gradient surrogate” as defined by Mairal (2013), because that line involves directly minimizing the second-order expansion of the loss function rather than an upper bound of the expansion. Hence, the correctness of Theorem 2 is in question.
> >
> >     Given Theorem 1, which states that the loss function is convex and smooth with respect to $\beta$, the issue with adopting the result of Mairal (2013) can be easily circumvented by adopting standard first-order methods such as projected gradient descent and accelerated gradient descent.
> >
> > 2. The constraint set $\Theta$ in (7) is not defined. This is related to Line 11 in the proposed optimization method (Algorithm 1) and the following remark on implementing the method:
> > > In this paper, we employ the L-BFGS(-B) algorithm (Byrd et al., 1995; Zhu et al., 1997), which is an iterative method for solving
> > non-linear optimization problems (with bounded constraints).
> >
> >     It is unclear why $\alpha$ should be a bounded parameter.
> > 3. The convergence guarantee only ensures convergence to a stationary point and does not guarantee global optimality, so the proposed statistical method still lacks a rigorous implementation. In addition, it would be beneficial to provide the rigorous definition of a stationary point.
> >
> > 4. Thank you for the clarifications about the proof of Theorem 5 (Theorem 4 in the previous version). Upon re-examination, I realized that this is a presentation issue and that my confusion stemmed from the writing. In particular, I believe the first paragraph of Appendix H does not precisely convey the message; the first sentence of the second paragraph is grammatically incorrect and hard to understand; there was a critical typo in the third sentence in the previous version, which prevented me from identifying the proposition it refers to. Please consider rewriting the proof.

---

> > > ### Author Response · Authors · 2024-09-06
> > > **Response to Reviewer rUk6**
> > >
> > > We sincerely appreciate your additional comments.
> > > In the revised paper, we have addressed four concerns you pointed out.
> > >
> > >
> > >
> > > ### **(Additional) Weakness 1**
> > > Thank you for your careful reading. As you suggested, our surrogate function did not completely correspond to a proximal gradient surrogate, which is defined by Mairal (2013). Our previous proof of Theorem 2 must be revised.
> > >
> > > We have modified our proof based on Definition 2.1 of Mairal (2013).
> > > Specifically, we have proven that our surrogate function satisfies majorization and smoothness properties and belongs to a first-order surrogate class. Please see Appendixes C and D of the revised paper for details.
> > >
> > > **Major Revisions:**
> > > - "Proximal gradient surrogate" and related descriptions have been deleted.
> > > - The derivation of our surrogate function has been rewritten in more detail in Appendix C.
> > > - A new proof of Theorem 2 has been added in Appendix D.
> > >
> > >
> > >
> > >
> > >
> > > ### **(Additional) Weakness 2**
> > > As shown on Line 11 in Algorithm 1, we optimize $\alpha$ by the L-BFGS algorithm without any bounded constraints since it is NOT a bounded parameter.
> > > On the other hand, as shown on Line 10 in Algorithm 1, we optimize $\sigma$ by the L-BFGS-B algorithm, which extends the L-BFGS algorithm to handle bounded constraints, since it is a bounded parameter such as $\sigma>0$.
> > >
> > > The sentence you quoted was intended to convey this. However, this expression was ambiguous and confusing. In the revised paper, we have improved this point by clarifying the differences between $\sigma$ and $\alpha$, as mentioned above. Moreover, the constraint set $\Theta$ in (7) has been deleted since it was also confusing.
> > >
> > > **Major Revisions:**
> > > - The constraint set $\Theta$ in (7) has been deleted.
> > > - We have improved the sentences in Section 2.3.
> > >
> > >
> > >
> > >
> > >
> > > ### **(Additional) Weakness 3**
> > > In non-convex optimization, the objective function can have multiple local minima and saddle points. This complexity makes it difficult to ensure that an optimization algorithm will converge to the global minimum rather than getting trapped in a local minimum. For this reason, in non-convex optimization, the theoretical goal is usually to guarantee the first-order stationary condition instead of global optimality (Jain & Kar, 2017; Lan, 2020).
> > >
> > > In non-convex optimization, the (block) coordinate descent often fails to converge to a stationary point (e.g., Powell's example; see Section 3.1 of Wright (2015) for details.) This is because the optimal solution could oscillate, diverge, or get stuck at a non-stationary point due to successively minimizing along coordinate directions. Theorem 2 guarantees that such situations do not occur and that Algorithm 1 converges to a stationary point.
> > >
> > > In response to your helpful comment, we have added the definition of a stationary point in Section 2.3. A stationary point here is defined as a point where the directional derivative in any direction is non-negative. This definition is well-used in non-convex optimization. Please see Section 2.3 of the revised paper for details.
> > >
> > > **References:**
> > > - Prateek Jain and Purushottam Kar. Non-convex optimization for machine learning. *Foundations and Trends in Machine Learning*, 10(3-4):142-363, 2017.
> > > - Guanghui Lan. First-order and stochastic optimization methods for machine learning. *Cham: Springer*, 2020.
> > > - Stephen J Wright. Coordinate descent algorithms. *Mathematical programming*, 151(1):3-34, 2015.
> > >
> > > **Major Revisions:**
> > > - The definition of a stationary point has been added just before Theorem 2 in Section 2.3.
> > >
> > >
> > >
> > >
> > >
> > > ### **(Additional) Weakness 4**
> > > As you pointed out, our first submitted paper contained a critical typo, which was caused by a Latex command error. We guess that you could not get to the correct proposition number because of this mistake. We apologize for your inconvenience.
> > >
> > > In response to your comments, we have rewritten the proof of Theorem 5 in Appendix H and tried to clarify it. Please see Appendix H of the revised paper for details.
> > >
> > > **Major Revisions:**
> > > - We have rewritten Appendix H.

---

> > > > ### Comment · Reviewer_rUk6 · 2024-09-06
> > > >
> > > > Thanks for the response!

---

### Review · Reviewer_7Eky · 2024-05-29

**Summary Of Contributions:**

This paper proposes to minimize the log-probability of a skewed distribution in order to fit a sparse linear regression model with skewed noise

**Audience:**

Yes

**Claims And Evidence:**

No

**Requested Changes:**

See the two weakness. I believe the authors need to benchmark the proposed optimization method against known ones or remove any claim of contribution based on it

**Strengths And Weaknesses:**

S: the problem where the noise in the data is skewed is both interesting and practically relevant. The method allows finding the level of skewedness in the data, which is a plus

W: I think the main weakness is the optimization proposal lacks motivation and needs additional comparison.
W1. Motivation: the main motivation is stated as "the solution could be unstable in optimization due to the non-convexity
 of the Lasso-type problem." This is not true. See https://arxiv.org/abs/2210.01212, where the authors proved that L1 regularization can be solved efficiently for generic nonlinear objective functions with simple gradient descent

W2. This is related to W1. I believe that the authors should avoid reinventing the wheel. As generic nonlinear solvers of L1 exists, the authors should either (1) rely on existing optimization method or (2) propose a new method and show that it has advantage over existing methods.

So far, the manuscript only proposes an optimization method without any motivation for the need of it, and I think this point needs to be improved

---

> ### Author Response · Authors · 2024-06-24
> **Response to Reviewer 7Eky**
>
> We appreciate your comments. As you pointed out, we may not have adequately explained the motivation behind our optimization method. In the revised paper, we have tried clarifying the motivation.
> Moreover, according to your request, we have also removed the sentence: "The solution could be unstable in optimization due to the non-convexity of the Lasso-type problem."
>
>
>
> ### **Weakness 1**
> Thank you for introducing Liu and Wang (2023) (https://arxiv.org/abs/2210.01212). Liu and Wang (2023) proposed a new optimization method using a reparametrization trick for the $\ell_1$-regularized parameter under the condition that *the optimization for the non-regularized parameter is easily obtained*. In particular, the non-regularized parameter is implicitly assumed to attain a global minimum using other general optimization methods in all theorems of Liu and Wang (2023).
>
> However, our situation seems to be different. The $\ell_1$-regularized parameter $\beta$ is easily optimized without reparametrization since our Theorem 1 shows its convexity (or strongly convexity). In contrast, the non-regularized parameters $\sigma$ and $\alpha$ are not easily optimized due to non-convexicity. For this reason, unfortunately, their idea cannot be applied to our situation.
>
>
>
> ### **Weakness 2**
> Following Reviewer rUk6's comment, we have obtained a new theorem about the convergence of our proposed algorithm. As you pointed out, many types of algorithms are applicable to our problem. However, our proposed algorithm has a convergence guarantee.

---

### Review · Reviewer_j3Vk · 2024-06-28

**Summary Of Contributions:**

In this manuscript the authors formulate a linear regression problem where the noise distribution is skewed and "mode-invariant", meaning that for any choice of the distribution parameters, the location parameter is equal to the mode. The authors present a novel alternating minimization method for solving this problem; one of the minimization problems is convex, while the other is non-convex but only involves two scalar variables. The authors prove theoretical guarantees on the excess risk, sparsity, and likelihood of their proposed solution, under certain assumptions. Finally, the authors demonstrate the improved performance of their method on simulated data -- including data simulated under model mis-specification -- and on two real datasets.

**Audience:**

Yes

**Claims And Evidence:**

Yes

**Requested Changes:**

Below I elaborate on the above weaknesses and what I would like to see.

[Critical to securing recommendation.] Improving writing.

The writing of the manuscript can often be unclear, with definitions sometimes not given and non-standard terminology used. Below I list some specific examples that I would like to see updated.

- Abstract: authors note that “regression model is always for a mode”. I have not heard this terminology before and it is imprecise. I suggest the authors change this to instead note that the noise distribution  always  has an interpretable mode parameter.
    - e.g. similarly, in the introduction, the authors note that “Chen et al treated a regression model on…” which is also imprecise terminology.
    - Sec 2.4: “that regresses the location parameter” also is imprecise.
- Figure 1: label the blue/black lines in the caption.
- Some variables are not defined and some equations are not clearly explained.
    - $f_{\psi}$ is not defined in (13)
    - Sec 3.2: authors should explicitly state what $\psi^0$ and $\psi$ are. Is $\psi^0$ the true model?
    - Sec 3.3: “has the order slightly larger than…”. I am not sure what “has the order” means. I suspect the precise terminology you want is that the excess risk/sparsity of your method is (log N)^2 larger than that of ordinary Lasso?
        - Same comment for Sec 3.4.
    - Eq (58): $v_n$ is introduced without definition. I assume it is equal to $r_{\alpha}(u_n)$?
    - Moreover, eq (59) does not immediately follow from Prop 5. The authors should write out this argument more clearly.

[Would strengthen manuscript]

- Theory could be validated empirically.

The authors make several assumptions in their convergence proofs, e.g. the restricted eigenvalue condition, and also present theoretical results on the quadratic margin (Theorem 3) and convergence (Theorem 4). It would be useful to see these results validated empirically, e.g. how loose/tight are the bounds derived by the authors?

- Real data evaluation is weak.

While the real data evaluation is interesting, the authors only perform a shallow comparison of their method versus others by looking at summary statistics. I suggest the authors compare the results of different models in more detail. For example, the authors should compare which coefficients are found by the authors approach that are not found by others, and vice-versa. Are the coefficients uniquely found by the authors’ approach more medically plausible? Do Lasso/other approaches find many false positives?

**Strengths And Weaknesses:**

Strengths:

- Novel and useful problem formulation. I believe the problem of regression with a skewed noise distribution is an important problem, and it is quite surprising that regression with a mode-invariant noise distribution has not been studied.
- The authors’ proposed alternating minimization method is well-motivated.
- Theoretical guarantees are interesting and non-trivial. In particular, I think Theorem 3 is a nice result.
- Simulated results adequately demonstrate the improved performance of their method, particularly in the presence of model mis-specification.

Weaknesses (see below for more comments):

- Writing is somewhat unclear, e.g. definitions for variables are missing, language can be confusing.
- Theory could be validated empirically.
- Real data evaluation could be investigated more thoroughly.

---

> ### Author Response · Authors · 2024-07-10
> **Response to Reviewer j3Vk**
>
> We sincerely appreciate your many suggestions. Your constructive comments have significantly improved the quality and clarity of this work. We have addressed the three weaknesses you pointed out as follows.
>
>
>
> ### **Weakness 1**
> We have made all required changes as described below.
>
> **Major Revisions:**
> - Imprecise terminology: We have tried to improve all similar expressions in addition to the abstract, Section 1, and Section 2.4 that you pointed out.
> - Figure 1: Explanations for the blue and black (and red) dashed lines have been added in the caption of Figure 1.
> - $f_{\psi}$ in (12): The definition has been added just before (12).
> - $\psi^0$ in Section 3.2: $\psi^0$ is the true model. The definition has been added in Assumption 1.
> - What "has the order" means: Following your suggestion, we have improved the sentences in Sections 3.3 and 3.4.
> - $v_n$ in (58): $v_n$ is equal to $r_{\alpha}(u_n)$. The definition has been added just before (58).
> - Derivation of (59): We have given an additional explanation for deriving (59) from Proposition 5.
>
>
>
> ### **Weakness 2**
> Your suggestion to validate our theoretical results empirically is highly intriguing. However, we were unable to thoroughly validate our theories and assumptions through numerical experiments due to time constraints. We would like to share our perspectives on your question below.
>
> **Assumptions 1 and 2:**
> Assumption 1 is just the bounded conditions of the feature vector and parameter space. This is often used in theoretical analysis. Assumption 2 (restricted eigenvalue condition) is often used in non-asymptotic analysis of $\ell_1$-penalty method. This is assumed on the covariance matrix of the feature vector and is independent of the skew-unomodal distribution we treat. Therefore, we think empirical examinations for the assumptions would not be necessary in this paper.
>
>
> **Theorems 3 and 4:**
> In Theorem 3 (quadratic margin), the constant $C$ appears in the lower bound. This constant $C$ also appears in the estimation error bound of Theorem 4. The constant $C$ is the key of these theorems when we consider how tights our bounds are in Theorems 3 and 4. First, we consider Theorem 3. In the case of the normal noise, the quadratic margin can be tight by using an appropriate constant $C$. In the case of skew-unimodal noise, as the skew parameter is larger, we think the bound will be looser. Next, we consider Theorem 4. The convergence rate of the estimation error bound is a standard one. We think the convergence rate will be tight, like the standard one.
>
>
>
> ### **Weakness 3**
> In the revised paper, we have also applied the proposed method to more easily interpretable data, specifically the Engineering Graduate Salary (EGS) prediction data (Aggarwal et al., 2016). This dataset has fewer features than the two medical datasets in Section 4.3, and the meanings of all features are specifically provided. The result shows that the proposed method outperformed the comparative methods and could select more reasonable features. Please see the revised paper for details.
>
> **Major Revisions:**
> - New experimental results for EGS have been added to Appendix I.3.
>
> **References:**
> - Varun Aggarwal, Shashank Srikant, and Harsh Nisar. Ameo 2015: A dataset comprising amcat test scores, biodata details and employment outcomes of job seekers. In *Proceedings of the 3rd IKDD Conference on Data Science, 2016*, pp. 1–2, 2016.

---

> > ### Comment · Reviewer_j3Vk · 2024-07-14
> >
> > Great! Your manuscript updates, particular the writing fixes and the new EGS results, have greatly improved the paper.

---

### Comment · Action_Editor_mCW2 · 2024-06-04
**Late review**

Dear reviewer nrra,

You have promised "to submit a review for this submission by the end of day on May 25, 2024 UTC time". The review is still missing now and we look forward to it to start the rebuttal. Thank you in advance!

AE

---

### Decision · Action_Editor_mCW2 · 2024-08-19

**Recommendation:** Accept as is

**Comment:**

The paper worked on a difficult regression problem with a linear regression model but the noise can be highly skewed (showed in Eq. (5)). It was well motivated to adopt a certain mode-invariant distribution as the model of the noise (showed in Fig. 1). There were also some theoretical guarantees for the proposed method. After the rebuttal, all the three reviewers were positive, and thus I think we can accept the paper for publication. Please incorporate your rebuttal into the camera-ready version.

**Audience:**

Yes

**Claims And Evidence:**

Yes

---

> ### Author Response · Authors · 2024-09-06
>
> Dear Action Editor and Reviewrs
>
> Once again, we appreciate your valuable feedback, which has enhanced our manuscript.
> We would like to inform you that the camera-ready version of our manuscript has been uploaded.
>
> Best regards,
>
> Authors